# Abiotic Stress in Crop Production

**DOI:** 10.3390/ijms24076603

**Published:** 2023-04-01

**Authors:** Romana Kopecká, Michaela Kameniarová, Martin Černý, Břetislav Brzobohatý, Jan Novák

**Affiliations:** Department of Molecular Biology and Radiobiology, Faculty of AgriSciences, Mendel University in Brno, 61300 Brno, Czech Republic

**Keywords:** abiotic stress, cold stress, crop, drought, heat stress, metabolites, phytohormones, salinity

## Abstract

The vast majority of agricultural land undergoes abiotic stress that can significantly reduce agricultural yields. Understanding the mechanisms of plant defenses against stresses and putting this knowledge into practice is, therefore, an integral part of sustainable agriculture. In this review, we focus on current findings in plant resistance to four cardinal abiotic stressors—drought, heat, salinity, and low temperatures. Apart from the description of the newly discovered mechanisms of signaling and resistance to abiotic stress, this review also focuses on the importance of primary and secondary metabolites, including carbohydrates, amino acids, phenolics, and phytohormones. A meta-analysis of transcriptomic studies concerning the model plant *Arabidopsis* demonstrates the long-observed phenomenon that abiotic stressors induce different signals and effects at the level of gene expression, but genes whose regulation is similar under most stressors can still be traced. The analysis further reveals the transcriptional modulation of Golgi-targeted proteins in response to heat stress. Our analysis also highlights several genes that are similarly regulated under all stress conditions. These genes support the central role of phytohormones in the abiotic stress response, and the importance of some of these in plant resistance has not yet been studied. Finally, this review provides information about the response to abiotic stress in major European crop plants—wheat, sugar beet, maize, potatoes, barley, sunflowers, grapes, rapeseed, tomatoes, and apples.

## 1. Introduction

Abiotic stressors, such as drought, soil salinity, heat, and cold, are major limiting factors affecting crop production both qualitatively and quantitatively [1]. These threats are likely to become even more significant under climate change and the pressures of an ever-growing human population. Recently, the global human population reached 8 billion people and the latest projections by the United Nations suggest that the world’s population could grow to around 8.5 billion by 2030 and 9.7 billion by 2050 [2]. Environmental change is, therefore, a big challenge for agriculture and its efforts to meet the growing need for food worldwide. Unfortunately, the vast majority of land is exposed to stressful conditions [3]. Compared to record yields, abiotic stress can reduce yields by more than 60% on average [4]. From a global perspective, climate extremes exhibit an increasing poleward gradient, and temperature variability patterns demonstrate the growing prevalence of heat extremes over cold ones [5]. A recent study based on sub-national yield data and a machine learning algorithm showed that climate extremes could explain up to half of the global crop yield variability [6]. The study also suggested that under the condition of regular irrigation, the yield anomalies are associated more strongly with temperature-related extremes than precipitation-related factors.

In this review, we focus on the European region, which, although medium in size, encompasses all the important climatic zones, including arid regions in the south and polar regions in the north [7]. According to FAOSTAT [8], in 2021, Europe was the largest producer of important crops such as barley (*Hordeum vulgare*), grapes (*Vitis vinifera*), and sugar beet (*Beta vulgaris*), and the second largest producer of wheat (*Triticum aestivum*), tomatoes (*Solanum lycopersicum*), potatoes (*Solanum tuberosum*), cucumbers (*Cucumis sativus*), and apples (*Malus domestica*).

The effects of climate change in Europe are regionally differentiated (Figure 1). In the Mediterranean region, temperatures are rising faster than in other parts of Europe and this increase is accompanied by increasing water demands and the risk of forest fires during the summer [9]. Desertification and a decrease in crop yields are serious threats to this area. In central and eastern Europe, summer precipitation is decreasing and the number of warm temperature extremes is rising [10]. Temperate and boreal vegetation over middle latitudes suffer from serious damage caused by accelerated phenological events overlapping with late-spring frosts [11]. Northern Europe is also affected by rising temperatures, but in this case, it can lead to increased crop yields through the introduction of new crop varieties, longer growing seasons, and the expansion of areas suitable for crop production [12,13]. A recent bioinformatic analysis of European crop losses over the last few decades showed that drought and heat waves were associated more with yield loss in cereals (9% and 7.3%, respectively) than in non-cereals (3.8% and 3.1%, respectively). The effect of cold waves was almost five times smaller [14].

The need to understand the mechanisms of resistance to abiotic stress and sustainable agriculture under a changing climate is reflected in the increasing number of publications focusing on crops and abiotic stressors (Figure 1B).

Crops require different conditions for optimal growth and to achieve marketable quality and these conditions govern where they are best cultivated [16]. The ten most commonly grown crops in Europe and the countries most involved in their production are shown in Figure 2A. In order of their level of production in 2021, the most cultivated crops are wheat, sugar beet, maize (*Zea mays*), potatoes, barley, sunflowers (*Helianthus annuus*), grapes, rapeseed (*Brassica napus*), tomatoes, and apples. Wheat is one of the main crops cultivated not only in Europe (Figure 2A) but also worldwide. It can adapt to a wide range of temperatures and environmental conditions, but excessive rainfall, together with high temperatures, can cause the spread of some common diseases that lower yields [17,18]. Water deficiency and high soil salinity also constrain wheat growth and development [19,20]. Another important cereal, maize, which has high water requirements, is susceptible to drought, particularly at certain critical stages of growth such as at the seedling or reproductive stage [21]. Sugar beet is a fairly salt-tolerant crop due to a number of mechanisms that help it to regulate the distribution of salt and other solutes within its tissues and maintain its water content [22]. In contrast, potatoes are one of the most drought-sensitive crops and high levels of salt in the soil adversely affect tuber development [23,24]. Barley is perhaps the most salt-tolerant cereal and exhibits a high level of resistance to abiotic stress [25]. Its high natural adaptability to a variety of growing conditions makes barley a promising crop for future production in a changing climate. Sunflowers are the second most important oil crop in Europe. Due to their relatively high resistance to abiotic stresses, they are often grown in semiarid and arid conditions. Nevertheless, their growth and production are still limited in drought and salt-stressed conditions [26,27]. Tomatoes can grow in almost all climatic regions of the world; however, their production is affected by drought, high salinity, and temperature extremes [28,29,30]. Apple fruit production is endangered by late-spring frosts that harm flower buds [31]. Rapeseed (oilseed rape or canola) is the principal oil crop in Europe and like other temperate region crops, it is susceptible to multiple abiotic stresses. The seed yield of rapeseed can be reduced more by heat stress than by drought alone [32].

In this review, we focus on various aspects of four important abiotic stressors that affect crop plants: drought, salinity, heat, and cold. Recent advances in the understanding of stress signaling and resistance in crop plants are summarized. We include a meta-analysis of gene expression in response to abiotic stress in the model plant *Arabidopsis.* This analysis revealed uncharacterized genes that are responsive to all abiotic stressors, as well as the transcriptional modulation of Golgi-targeted proteins in response to heat stress, which has not previously been well described.

## 2. Abiotic Stresses and Crops

### 2.1. Heat Stress

#### 2.1.1. Europe Is Experiencing the Hottest Summers in Recorded History

According to the Intergovernmental Panel on Climate Change, the temperature increase is likely to reach 1.5 °C between 2030 and 2052 if it continues to increase at the current rate [33]. In Europe, 2020 was the warmest year on record at 2.16 °C above average. Last year (2022), the average annual temperature was the second highest on record, while summer was the hottest. Moreover, the average rate of temperature increase in Europe has almost tripled since 1981 [34]. The areas most affected by high temperatures were in western and northern Europe and several countries experienced their highest summer wildfire emissions for at least the last 20 years [35].

#### 2.1.2. The Temperature Optimum for Most Crops Grown in Europe Is Exceeded for Weeks during the Season

Temperature optima differ according to the crop species, cultivar, and stage of development [36]. Based on the temperature optima for the most important European crops (Table 1), it can be concluded that temperatures above 30 °C are no longer optimal for European crop production. We searched for the number of days exceeding this threshold in the less heat-exposed regions of central Europe in 2022. For example, data from the Czech Hydrometeorological Institute (Doksany weather station) and the German Weather Service showed that 30 °C was exceeded on 35 days [15], and on an average of 17.3 days [15], respectively. These data show that during a year, crop plants are exposed to superoptimal temperature conditions, not for just a few days but rather for weeks, even in the colder European regions.

#### 2.1.3. Role of High Temperatures in Crop Production

Heat stress is defined as exposure to temperatures above the optimum that are sufficient to cause irreversible damage to plant growth and development [47]. Higher temperatures directly affect plant vegetative stages, resource allocation, and, above all, reproductive processes, which can lead to a substantial reduction in yields [48]. Extremely high temperatures result in rapid cellular injury and cell death [49]. Heat stress in the vegetative phase of a plant’s growth can lead to a lower rate of photosynthesis followed by biomass reduction [50]. Higher temperatures significantly affect the reproductive phases of development. For species of the genus *Brassica*, temperatures exceeding 29.5 °C during the period from bolting to the end of flowering reduce flowers, pod numbers, seed weight, and thus, total yield [51]. Aiqing et al. [52] stated that heat exposure during spring wheat gametogenesis is a major determining factor for yield loss. Bheemanahalli et al. [53] observed a 2–93% reduction in pollen germination after heat-stress treatment on spring wheat genotypes. Maize yield is usually lowered by kernel abortion, which can be a result of the low utilization of soluble sugar resulting from the modulation of starch synthesis during heat and drought stress [54]. These authors also emphasized the detrimental effect of thermal stress on pollen viability. High temperatures during the whole grain-filling stage in maize resulted in a 28% decrease in grain weight [55]. Many authors have shown that crop plants are very sensitive to heat stress, especially during reproductive phases.

#### 2.1.4. High-Temperature Signaling in Model Plant and Crops

Several thermosensors and thermosensitive elements have been characterized in reaction to heat stress. Thermosensors are elements that meet three conditions: a change in their structure or activity is based on direct interaction with heat-stress conditions; this change leads to important signals or responses; and these processes lead to physiological and morphological alterations or responses to stress [56]. Following these conditions, changes in membrane fluidity and changes in protein conformation can be evaluated as thermosensors [57]. Changes in membrane fluidity induced by high temperatures lead to Ca^2+^ influx due to the activation of Ca^2+^-permeable channels belonging to the CYCLIC NUCLEOTIDE-GATED CHANNEL (CNGC) family [58]. Specific CNGCs have been shown to mediate thermotolerance in *Arabidopsis thaliana* and *Oryza sativa* [59,60]. Important components of Ca^2+^ signaling pathways are annexins. This multifunctional protein family is now known to be associated with heat-stress responses in wheat, barley, rye (*Secale cereale*), rice, and tomatoes [61,62,63,64,65]. A change in protein conformation in reaction to heat stress has been confirmed in specific photoreceptors. PHYTOCHROME B (PHYB) in *Arabidopsis* has been observed to participate in temperature perception through its temperature-dependent reversion from the active Pfr state to the inactive Pr state [66]. The reversion of phytochrome is followed by downstream components that can be better specified as thermoresponsive [57]. The most well-known group of thermoresponsive interacting partners is the phytochrome-interacting factor (PIF) family [67]. The importance of PIFs for thermomorphogenesis has been observed in *Arabidopsis* and members of this family have also been identified in wheat, tomatoes, and rice [68,69,70,71], whereas increased expression levels of PIFs under high temperatures have been confirmed in maize [72]. The activity of PIF4 can influence the function of the heat-responsive gene *EARLY FLOWERING 3* (*ELF3)* in *Arabidopsis*, barley, and wheat [73,74,75]. Moreover, ELF3 has recently been recognized as a thermosensor [73]. The complex interactions of ELF3, PIF4, and PHYB suggest their function as a PHYB-ELF3-PIF4 module that regulates the plant’s responses to environmental cues, with implications for plant biorhythms [76]. Other important signaling components are protein kinases, Reactive Oxygen Species (ROS), and transcription factors [77]. Protein kinases play an important role in ROS activity in response to heat. After ROS production, the Mitogen-Activated Protein Kinase (MAPK) signaling pathway is activated and induces the expression of transcription factors [78]. One group of activated transcription factors is HSFs (heat-shock factors), which can induce the expression of various heat-shock proteins (HSP) [79]. BcHSFA1, Sly-HSFA1a, ZmHSF05, TaHSFA2e, and OsHSFA2dI are associated with improved heat tolerance and have been identified as activators of HSP expression in rapeseed, tomatoes, maize, wheat, and rice [80,81,82,83,84]. The activity of HSFs can be regulated by heat-shock proteins (HSPs) [85] and alternative splicing [86]. The alternative splicing mechanism has been confirmed as a novel component of heat-shock memory in wheat, barley, rice, maize, and tomatoes [84,87,88,89,90]. Signaling mechanisms are summarized in Figure 3 in the red section.

### 2.2. Drought Stress

#### 2.2.1. High Temperatures without Precipitation Are Causing Drought across Europe

High temperatures are strongly associated with drought stress. Drought is a condition in which rain is either lacking or insufficient for so long that a considerable hydrological imbalance results. As a direct consequence of water shortage, plants, being sessile organisms, are affected by drought stress. Drought stress is becoming more significant with decreased availability and increasing demand for water. Agriculture presently accounts for 70% of global clean water demand and this percentage is expected to increase rapidly over the coming years [91]. According to the European Drought Observatory, drought is not limited only to some specific regions in Europe [35]. Insufficient rainfall, as indicated by the standardized precipitation index, was detected during the summer of 2022 in all main crop-producing countries in Europe, and it correlated with a significantly lower soil-moisture anomaly index across the whole of Europe.

#### 2.2.2. Impact of Drought on Different Crops at Different Developmental Stages

The effect of drought on yield depends on the severity of the drought stress. Severe drought causes plant death so yield loss without irrigation could be total, as has been shown for barley under field conditions [92]. A meta-study based on data from field experiments showed that the loss of yield when water was reduced by approximately 40% varied from around 21% in wheat to 39% in maize [93]. A 10% yield decrease has been predicted for wheat and maize as a result of drought in European countries [94]. Like other abiotic stresses, yield loss caused by drought varies with cultivars [92,95], and this factor should not be overlooked when species are compared regarding their drought resistance. Moreover, drought also impairs crop quality, e.g., lower seed oil content has been reported in seeds of rapeseed under water-shortage conditions [96]. A relevant factor in yield loss is also the plant developmental stage when affected by drought. Plants exposed to drought in the reproductive stage are more vulnerable to drought [93]; therefore, better timing of plant development could improve yields [97]. Prolonged exposure to drought stress during the reproductive phase decreases grain filling, flower production, seed composition, and longevity [98,99].

#### 2.2.3. Drought in Plant Physiology

The main problem associated with drought is the initiation of many intertwined positive feedback loops that exacerbate drought-stress conditions and lead to restrictions in above-ground growth. One such limitation is the closure of stomata as a mechanism to protect against water loss, which, in turn, limits the CO_2_ concentration. Limiting CO_2_ inhibits the productivity of the photosynthetic process and promotes the formation of ROS as a byproduct of the electron transport chain, without a sufficient level of terminal energy acceptors [100]. ROS, in turn, impair the photosynthetic apparatus and cause oxidation of other important molecules, including proteins and lipids. The transpiration ratio for CO_2_ fixation is around 400 in C3 plants [101]. However, the limitation of CO_2_ levels has serious drawbacks and the importance of maintaining sufficient CO_2_ levels has led to the evolutionary decoupling of light reactions and CO_2_ uptake in CAM plants. Another feature of stomata closure is the onset of leaf-tissue overheating. A decrease in the mass flow of water reduces nutrient acquisition and causes a loss of turgor, ultimately leading to a reduction in plant growth, leaf area, and leaf numbers, resulting in a reduced photosynthetically active area [102,103]. Plants are forced to synthesize osmoprotective compounds rather than investing their resources in optimal growth. Overall, drought stress affects diverse physiological processes, ultimately leading to growth restrictions and low production [102,103].

#### 2.2.4. The Role of Root Growth in Drought Resistance

The response to drought stress depends on its severity [104] but the main counteracting mechanisms involve improving water uptake, reducing water loss, and tolerating water deficiency. Water uptake is mainly driven by efficient root growth. Interestingly, breeding technologies in the past were more focused on above-ground organs neglecting root systems or acquiring their properties only implicitly or indirectly [105]. Thus, roots and their plasticity are a promising source of genetic variation for stress adaptation. Over the last decade, different groups have identified QTL, which influences root and physiological traits in important crops [105]. The generally favorable properties of the root system for drought conditions are deep and branched rooting and efficient water uptake. Plants with shallow roots such as Styrian pumpkin (*Cucurbita pepo* L. *Styriaca*) could have serious problems with water uptake in field conditions currently prevalent [106]. In this sense, the importance of root growth angle has been demonstrated by characterizing the *DEEPER ROOTING 1* (*OsDRO1*) gene in rice [107]. *OsDRO1* promotes deep rooting and maintains a high yield under drought conditions Recently, the homologous gene in maize *ZmDRO1* has been shown to modulate root angle, and its higher expression following stress signals has been associated with higher yields in field conditions [108]. However, *DRO1* ectopic expression without a stress signal could have a negative impact, even in non-stressed conditions, as has been shown for maize [108]. The positive effect of root architecture on drought resistance has also been demonstrated in *Arabidopsis* expressing the *StDRO1* gene from potatoes [109]. These reports suggest that identifying these genes in other crops such as wheat and barley [110] and breeding focused on root architecture could be fruitful strategies for preserving yields under drought conditions.

#### 2.2.5. Roots and Hydrotropism

Because water is not distributed homogeneously in the soil, growth toward regions of the soil with higher water potential could be crucial in water-limiting conditions. The signaling pathway employing the Ca^2+^ channel pattern along root cells, endoplasmic reticulum-localized type 2A Ca^2+^-ATPase (ECA1), and MIZU-KUSSEY1 (MIZ1), have been shown to play an important role in root hydrotropism in the model plant *Arabidopsis* [111], although the complete molecular mechanism, including the sensor, remains unclear. Part of the mechanism could be hyperosmolality-gated Ca^2+^-permeable ion channels such as OSCA1 that can modulate Ca^2+^ concentration in response to osmotic stress [112] and distinguish between osmotic and ionic stress [113]. Recently, the role of plasma membrane-localized OSCA1.1 in root bending and hydrotropism has also been confirmed [114]. As a promising candidate for drought resistance, OsOSCA1.2 hyperosmolality gating has been studied by cryogenic electron microscopy, and the characterization of *OSCA* genes has also been performed for barley, soybeans, and maize [115,116,117]. However, the detailed molecular mechanism of hydrotropism is still not completely known, even in the model plant *Arabidopsis*. We should emphasize that experiments explaining root responses must be planned with respect to the natural root environment because some artificial conditions, such as the illumination of roots, could interfere with hydrotropism [118] and affect other water-related root features such as root hair growth [119].

#### 2.2.6. Long-Distance Signaling of Water Deficit

Regulation of transpiration is the main process in reducing water loss. Here, the key mechanism is the hormonal regulation of stomata closure by abscisic acid (ABA) [120]. Part of the long-distance signaling component is the root-derived small peptide CLAVATA3/ESR (CLE)-RELATED PROTEIN 25 (CLE25). In response to drought, CLE25 moves from the roots to the leaves, where it is perceived by BARELY ANY MERISTEM (BAM) receptors and induces accumulation of ABA by activating the biosynthetic enzyme NINE-CIS-EPOXYCAROTENOID DIOXYGENASE 3 (NCED3) [121]. ABA has been established as a very necessary stress-related phytohormone through research focused on crop plants [122,123]. The role of ABA and its interaction with other plant hormones is discussed in Section 4.2 on metabolites. The signal for stomata closure is also mediated by ROS. In response to stress, H_2_O_2_ activates the receptor LEUCINE-RICH REPEAT RECEPTOR PROTEIN KINASE (HPCA1) to open the Ca^2+^ channels and close the stomata [124]. Mutants in *HPCA1* have also confirmed the role of this receptor in ABA downstream events, leading to stomata control. In contrast to *Arabidopsis*, ROS signaling in crops is far from being elucidated. In addition to stomata control, water loss can also be reduced through modifications to the cuticle, changes in leaf anatomy, or by utilizing the unique metabolism found in CAM plants [125,126,127]. The signaling mechanisms are summarized in the yellow section in Figure 3.

#### 2.2.7. Other Mechanisms of Drought Resistance

Tolerance to drought is based on counteracting the decrease in water potential by accumulating osmoprotectants, adjusting the metabolism, increasing water-usage effectiveness (WUE), and reducing the impact of the secondary effects of drought, including improving antioxidant metabolism and protecting photosynthesis. The role of metabolites in stress tolerance is discussed in more detail in a separate section.

### 2.3. Salt Stress

#### 2.3.1. Plants Are Variable in Their Resistance to Salt

Most crop plants are glycophytes that are sensitive to salinity stress. It has been estimated that approximately half of all irrigated land is affected by salinity [128]. Highly saline soil can result from natural processes (weathering, rain with sea-salt content) or human activity (land clearing, irrigation) (reviewed in [129]). The level of salinity is usually evaluated by the electrical conductivity of a saturated extract of soil or irrigation water. Salt-sensitive vegetables such as beans (*Phaseolus vulgaris*), carrots (*Daucus carota*), or onions (*Allium cepa*) can only tolerate low levels of salinity, with a threshold of detrimental effects around 1 dS/m, which is well below the level of 4 dS/m used to classify soils as saline [25]. Salt-tolerant plants include those with a threshold greater than 4 dS/m. A broad overview of the salt-stress thresholds in soil for different plant species was prepared and published by Utah State University Extension [130] and an extension, including thresholds for irrigation water, has also been reviewed [131]. The yield of a crop grown in soils of increasing salinity varies not only with plant species but also with cultivars, as has been demonstrated recently under laboratory conditions for olives (*Olea europaea*) [132] and maize [133] and in field conditions for wheat, barley [134], and potatoes [135]. Although searching for resistant cultivars or varieties is thus a promising strategy for maintaining yields where the salinity of the land is increasing, it is clearly not a final solution.

#### 2.3.2. Key Mechanism for Salt-Stress Resistance

Salt stress negatively affects plant growth, development, and production [136]. Increased levels of salt have two distinct biophysical consequences: induction of the osmotic stress and accumulation of ions to cytotoxic levels. This section focuses on the second of these consequences. One of the key results of excess salt is ion imbalance within the cell [137], as the homeostatic balance between ion ratios is thought to be the basic mechanism of tolerance to increased salt levels [138]. In plants, the Salt Overly Sensitive (SOS) pathway is a core mechanism for salt tolerance [139]. The principal determinant of Na^+^ extrusion from the cytoplasm to the apoplast is the cytoplasmic membrane-embedded Na^+^/H^+^ anti-porter SOS1 [140]. The activation of this antiporter in *Arabidopsis* is driven by two calcium sensors, CALCINEURIN B-LIKE PROTEIN 4 (AtCBL4/AtSOS3) and CALCINEURIN B-LIKE PROTEIN 10 (AtCBL10/AtSCABP8), and CBL-INTERACTING PROTEIN KINASE 24 (AtCIPK24/AtSOS2) [141]. The calcium sensors perceive the salt-induced [Ca^2+^]_cyt_ and promote SOS2 activity. Activated SOS2 is recruited to the plasma membrane to phosphorylate the AtSOS1 antiporter, which, in turn, prevents the accumulation of Na^+^ at toxic levels. Mutants in the SOS pathway exhibit higher sensitivity to salt treatment [140] and their overexpression has been shown to significantly increase salt tolerance [142]. Recently, orthologs of members of the SOS pathway have been identified and confirmed in crop plants. Like the model plant *Arabidopsis*, SOS1 mutants in rice or maize also exhibit salt hypersensitivity [143,144]. Members of the SOS pathway have also been confirmed in tomatoes [145,146,147] and maize [144], although, to date, the SOS pathway has not been fully described and confirmed in other crops.

#### 2.3.3. Salt-Stress Signaling and Role of ROS

The SOS pathway is dependent on the initial salt-induced increase in cytosolic calcium. It has recently been demonstrated that candidates for the Na^+^ sensor are glycosyl inositol phosporylceramide (GIPC) sphingolipids that are regulated by INOSITOL PHOSPHORYLCERAMIDE GLUCURONOSYLTRANSFERASE 1 (AtMOCA1/AtIPUT1) [148]. GIPC sphingolipids directly bind Na^+^ ions in the apoplast and regulate Ca^2+^ influx to the cell by an unknown ion channel, and their decreased levels in a *moca1* mutant were followed by lower [Ca^2+^]_cyt_ and salt hypersensitivity [148]. Liu et al. (2022) confirmed the importance of IbIPUT1 in sweet potatoes (*Ipomoea batatas*) [149] but its role is less known in other important crop plants.

Salt stress is tightly connected with ROS metabolism and signaling in cells [150,151]. ROS, along with the vacuolar ion channel TWO PORE CHANNEL1 (TPC1), assist with a salt-induced calcium wave in the plant body [152] that optimizes the response to salt stress by regulating gene expression before the onset of stress in distal shoot tissues [153]. ROS are generated by RESPIRATORY BURST OXIDASE HOMOLOGS (RBOH) in the response to salt treatment [152] and RBOH inhibition suppresses Na^+^ efflux from the cells [154]. Although stress-induced-extracellular H_2_O_2_ is perceived by HPCA1, leading to an increase in [Ca^2+^]_cyt_ [124], it is unlikely to play a primary role in the regulation of [Ca^2+^]_cyt_ in response to salt stress because both *hpca1* mutants and wild-type plants respond to NaCl treatment similarly [124]. ROS signaling plays an important role in salt tolerance but excess ROS beyond their signaling function significantly reduces tolerance to salt stress, as demonstrated in plants that possess an improved antioxidant system through the overexpression of antioxidant enzymes [151] or the foliar application of nanoparticles [155]. The signaling mechanisms are summarized in the green section in Figure 3.

### 2.4. Cold Stress

#### 2.4.1. Crop Sensitivity to Low Temperatures

Crops can be classified with respect to their frost resistance as chilling sensitive, chilling tolerant, or freezing tolerant [156]. Sensitivity to temperature thresholds varies according to crop species, cultivar, and an individual plant’s developmental stage [157,158]. Many important crops can build a cold tolerance but sudden temperature changes, the occurrence of which is increasing with climate change, do not allow for the process of acclimatization. The absence of low-temperature-induced cold-hardening caused by the warm weather prevalent during the fall/winter and reduced snow cover may expose the plants to unforeseen freezing conditions, which they cannot survive. If the temperature is not lethal, low temperatures can still influence the vegetative and reproductive growth of the plants. Cold stress can lead to poor germination, chlorosis, wilting, growth retardation, flower abscission, pollen sterility, or reduced fruit set [159,160,161,162]. Low temperatures affect not just the overall yield but also the seed and plant quality. A significant decrease in seed size (24%), starch (34%), protein (33%), and fat (43%) reserves caused by cold has been observed in chickpeas (*Cicer arietinum*), along with an increase in the level of soluble sugars [163]. The total content of grapevine phenolic compounds can decrease after prolonged cold stress [164]. Low temperatures reduce the amylose, amylopectin, and total starch concentrations in grains of wheat, whereas more drastic changes have been observed when low-temperature treatments occur during the booting stage rather than at the jointing stage [165].

Among the selected crops cultivated in Europe, wheat belongs to a freezing-tolerant species [166]. Potatoes are considered chilling-tolerant crops, whereas maize, tomatoes, and grapes are considered chilling-sensitive crops [167,168]. Sugar beet is a potential winter crop but low winter temperatures limit its production [169]. It has been shown that temperatures as low as −5 °C do not affect the survival of sugar beet plants, whereas at −7 °C, the plant survival rate decreases to 50% and temperatures from −9 °C to −15 °C completely kills them [170]. Even though wheat is known to tolerate freezing conditions, wheat production can be influenced by late-spring frosts and severe winter frosts without adequate snow cover [171].

#### 2.4.2. Low-Temperature Signaling

Low temperatures cause changes in the fluidity of the plant cell membrane that are followed by the activation of Ca^2+^ channels and receptor-like kinases in the plasma membrane [172]. These changes trigger a cascade of Ca^2+^ and MAPK signals, similar to the heat-stress-sensing mechanism [173]. In plants, there is no known exclusive protein that serves as a cold receptor but there are several candidates, including the G-protein signaling receptor COLD1, CNGCs, glutamate receptors, and PHYB [66,174,175,176]. In rice, COLD1 coupled with RICE G-PROTEIN α SUBUNIT1 (OsRGA1, not to be confused with *Arabidopsis* DELLA protein AtRGA1) is involved in cold sensing by modulating calcium signals [174]. The expression of CNGCs increases after exposure to cold in rice, tobacco (*Nicotiana tabacum*), and *Brassica oleracea* [175,177,178]. CNGCs have been described as thermosensors both in *Arabidopsis* and moss [179] but their molecular mechanisms are still not fully understood. In rice, *cngc14* and *cngc16* mutants display reduced survival rates and a higher accumulation of hydrogen peroxide after exposure to heat or chilling stress, which indicates a critical role of CNGC genes under both conditions [60]. Glutamate receptors AtGLR1.2 and 1.3 positively enhance cold tolerance in *Arabidopsis* by promoting jasmonate accumulation in response to cold [176]. The perception of low temperatures initializes the ICE-CBF-COR signaling pathway, which consists of an INDUCER OF CBF EXPRESSION (ICE), C-REPEAT BINDING FACTOR (CBFs), and COLD-RESPONSIVE GENES (*CORs*). ICE1 is an MYC-like basic helix–loop–helix transcription factor that binds to the MYC cis-acting elements in the *CBF* promoter and positively regulates its function [180]. The function of ICE1 depends on its post-translational modification by HIGH EXPRESSION OF OSMOTICALLY RESPONSIVE GENES 1 (HOS1) [181]. In maize, ICE1 not only regulates the expression of *CBFs* directly but also changes amino-acid metabolism and thus regulates mitochondrial oxidative bursts that impair cold tolerance [182]. CBFs are described as master transcription factors that regulate the expression of approximately 12% of *COR* genes, whose products function in the cold acclimation process and the acquisition of freezing tolerance [183,184]. Proteins involved in cold adaptation include dehydrins, Late Embryogenesis Abundant (LEA) proteins, antifreeze proteins, ROS detoxifiers, enzymes of osmoprotectants biosynthesis, lipid metabolism proteins, chloroplast proteins, and the other metabolites described in a separate section. In *Arabidopsis*, the three tandemly arranged *CBF* genes, *CBF1*, *CBF2*, and *CBF3*, are involved in cold acclimation and their expression is induced within 15 min of exposure to low temperatures [185]. Despite the fact that freezing-sensitive tomatoes are unable to cold-acclimate, they encode three CBF homologs; however, only one of them, *SlCBF1*, is cold-inducible. Even so, the overexpression of tomato *LeCBF1* in transgenic *Arabidopsis* increases its freezing tolerance [186]. CBF homologs have been found in many other crop species such as wheat, barley, maize, lettuce, and apples [187,188,189,190,191]. The involvement of *TaCBF14* and *TaCBF15* from winter wheat in the cold acclimation process has been demonstrated by the overexpression of these genes in spring barley, where the transgenes improved the frost tolerance of barley by several degrees [192].

#### 2.4.3. Role of Redox Changes in Cold Signaling

Recently, H_2_O_2_ has been shown to activate plant cold responses through its effect on the sulfenylation of cytosolic ENOLASE 2 (ENO2). In response to H_2_O_2_, ENO2 forms oligomers that are imported into the nucleus, where they activate *CBF* expression [193]. Another described mechanism by which temperatures regulate the activity of CBFs is through cold-mediated redox changes that induce the structural switching and functional activation of CBFs. After exposure to cold, THIOREDOXIN H2 (Trx-h2), which is anchored to cytoplasmic endomembranes through myristoylation, is released and translocates to the nucleus, where it interacts with CBF1. Trx-h2 reduces the oxidized CBF proteins and switches them to an active state to regulate downstream targets [194]. The importance of Trx-h2 in cold tolerance has been demonstrated in *Citrullus lanatus* [195] but the role of redox regulation in cold signaling in other crops is still not fully understood. The signaling mechanisms are summarized in the blue section in Figure 3.

## 3. Similarities among Abiotic Stresses and Their Potential Crosstalk

### 3.1. Meta-Analysis of Stress-Responsive Genes

Abiotic stresses activate specific molecules and mechanisms in plant organisms. Plants often experience different combinations of stresses, which can result in interactions between specific signaling pathways [196]. A strong effect of combined abiotic stresses on yield parameters has been observed in wheat, maize, and barley [197,198,199]. The fresh weight of maize shoots was reduced by 48% (heat-drought), 11% (heat), and 24% (drought) after stress treatment of the FH-988 genotype. The same attribute was reduced by 19% (heat-drought), 13% (heat), and 18% (drought) after stress treatment of the NT-6621 maize genotype [198]. Combined heat and drought stress in barley led to a reduction in yields by more than 95% in all tested varieties [199]. Multiple stress responses could lead to the activation of pathways and specific effects that are still not fully understood. The identification of specific target genes that occur in multiple abiotic stresses could provide a new perspective on this topic.

Meta-analyses are efficient approaches to the identification of important genes in abiotic stress responses. Recently, these bioinformatic analyses uncovered important abiotic stress targets in soya, wheat, and rice [200,201,202]. The results of the transcriptomic studies, which are available in the Expression Atlas database, provide a rich resource that allows analyses and comparisons of the effects of different stressors [203]. Here, we focus on the common regulation of genes significantly modulated by drought (or low water potential), heat, cold, or salinity. A total of 10,317 genes were found to be regulated by at least one stress condition, which at least doubled a gene’s expression (Figure 4).

In total, heat regulated 5992 genes (2462 upregulated, 3530 downregulated), cold regulated 4473 genes (2182 upregulated, 2291 downregulated), drought regulated 3875 genes (1509 upregulated, 2366 downregulated), and salt regulated 1857 genes (1143 upregulated, 714 downregulated). The weakest response was observed for the salt-stress treatment. Moreover, salt-responsive genes showed the most regulation compatible with other abiotic stresses. The highest overlap that reflected a similar pattern of expression was seen for salt stress and drought, with 89% of 211 regulated genes. Similar patterns were also observed for common genes with temperature stresses. The least compatible, in terms of gene expression, was the cold treatment, with approximately 50% of genes co-regulated in the same way as heat stress, and surprisingly, also drought stress. Of the total number of 190 genes regulated by all four stresses, only a small fraction (38 upregulated and 11 downregulated) was affected in the same manner.

### 3.2. Subcellular Localization of Products of Genes Involved in Abiotic Stress Response

Plants respond to unfavorable conditions by changing the patterns of protein expression. Stress signals are usually first recognized by the plasma membrane, where most receptor proteins are localized [215]. Mitochondria and chloroplasts are the primary sites for the production of ROS, and abiotic stress causes an imbalance between ROS and their scavengers, which negatively impacts the cell environment [216]. ROS also act as a signal that is transduced through cellular compartments and regulates gene and protein expression levels [217]. Since different subcellular elements play distinct roles in stress responses, the predicted subcellular localization of the stress-regulated genes’ products was subsequently analyzed (Figure 5A). The relative number of high-confidence marker proteins of genes regulated by abiotic stress was obtained from the SUBA5 (Figure 5A; Arabidopsis Subcellular Database; [218]), and the database provided the predicted localizations for half of the products of regulated genes.

The most obvious change was seen in mitochondria after temperature stress. Both heat and cold stress showed a similar pattern, with a highly enriched group of upregulated genes and a small group of downregulated genes targeted to the mitochondria. A similar pattern was observed for salinity and drought but here it was less apparent. Interestingly, genes of plastid localized proteins showed different patterns compared to those of mitochondria, with a higher fraction of downregulated genes, especially after cold and salinity treatment. A very specific response to stress was found for Golgi-localized proteins that were downregulated mainly under heat-stress conditions (Figure 5B). Less is known concerning the role of the Golgi than, for example, the role of hormones, either in crops or in *Arabidopsis*. In total, products of about 90 genes responsive to stresses were localized to the Golgi, 55 of which were downregulated by heat stress, suggesting an exceptional interaction between heat and the Golgi. The main process represented by these genes includes vesicular transportation, methylation, and pectin biosynthesis related to cell-wall organization. Pectin metabolism has been shown to play a role in heat tolerance in *Arabidopsis* and rice [219,220]. Drought and salinity partially overlap in their biophysical effect on plant cells by inducing osmotic stress, and both stresses show similar patterns in terms of cytosolic and extracellular responses. However, they differ significantly in the downregulation of mitochondrial and plastid proteins. Overall, the analysis showed different signatures of specific abiotic stresses.

**Figure 5 ijms-24-06603-f005:**
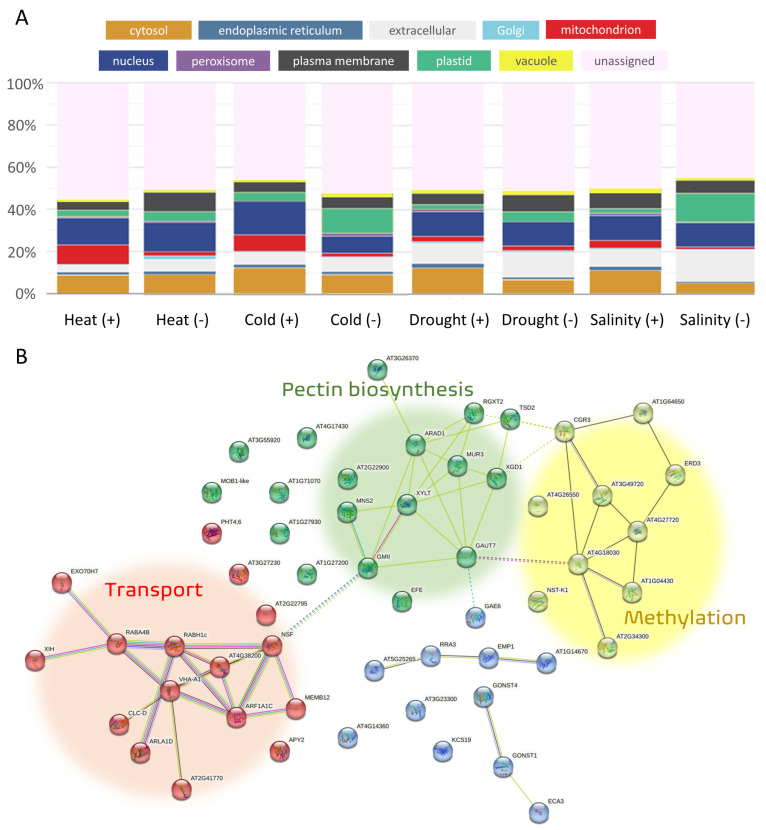
Relative predicted subcellular distribution of stress-responsive gene products calculated for each compartment using the online tool SUBA5 [218]. Data for the analysis were obtained from the Expression Atlas database [203]. (**A**) The genes upregulated by a specific stress are marked as (+) and those downregulated by a specific stress are marked as (-). The bars represent the subcellular distribution of stress-responsive proteins based on the AGI list of stress-regulated genes compared to the HCM list; hits are identified and summed per compartment. (**B**) Heat-stress downregulated genes with localization to the Golgi and their function. The interactions were visualized by String [221].

### 3.3. Functions of Common Genes Responding Identically to at Least Three Different Abiotic Stresses

Based on the results of the above analysis, 419 genes were found to be affected by at least 3 different stresses that are associated with significant processes in the plant’s abiotic stress response, and these genes are regulated in a similar manner (Figure 6, Appendix A).

The most represented groups in both upregulated and downregulated genes were cofactor biosynthesis, amino acid metabolism, transcription factors, transport, plant hormone signal transduction, and other enzymes. Categories specific to upregulated genes were carbohydrate metabolism and protein processing in the endoplasmic reticulum. The importance of carbohydrate metabolism for abiotic stress responses has been observed in durum wheat, barley, and tomatoes [223,224,225]. The endoplasmic reticulum plays an important role in the reprogramming transcription and translation of stress response regulators [226]. For example, the defense mechanisms of wheat-seedling leaves under salt stress are known to be associated with the endoplasmic reticulum [227], and SlbZIP60 splicing has been observed as part of the endoplasmic reticulum stress pathway in response to heat and virus infection in tomatoes [228]. Endoplasmic reticulum stress signaling has also been associated with drought tolerance in maize [229]. Categories specific to downregulated genes were purine metabolism, lipid, steroid metabolism, and histones. Drought-tolerant spring-wheat cultivars affected by drought stress exhibit significant changes in purine metabolism [230]. The intermediate product of the purine catabolic pathway, allantoin, has been found to accumulate in higher amounts in the leaves of drought-tolerant genotypes of rice [231]. Maize hybrid ND476 may save energy during drought stress by reducing purine and sterol metabolism [232]. Drought stress in rapeseed has been shown to result in changes in lipid metabolism, specifically a decrease in leaf polar lipids [233]. Changes in membrane lipid metabolism in maize leaves under cold stress were observed by Gu et al. [234]. Lipid alterations during heat stress have also been observed in wheat and tomatoes [235,236]. Histone modifications altering the responses to salt and drought stress have been observed in rice [237,238]. Tomato HSFB1 with a histone-like motif has also been observed to function as a transcription coactivator [239].

### 3.4. Specific Universal Stress Responsive Genes Affected by Heat, Cold, Drought, and Salinity

A total of 49 genes were found to be affected by all 4 stresses and were associated with various significant processes in plant abiotic stress responses (Figure 7).

In terms of regulation changes, 38 and 11 genes were identified as upregulated and downregulated, respectively. By demonstrating the well-known gene-specific transcriptional regulators associated with the stress response, the analysis supported the idea that their stress response is universal.

The first example is AT5G05410, also known as a DEHYDRATION-RESPONSIVE ELEMENT BINDING PROTEIN 2 (DREB2A). The overexpression of its ortholog *OsDREB2A* in transgenic rice plants revealed significant tolerance to osmotic, salt, and dehydration stress [241]. *ZmDREB2.9-S*, *ZmDREB2.2*, and *ZmDREB2.1/2A* were upregulated in response to cold, drought, and abscisic acid and may play redundant roles in stress resistance in maize [242]. The overexpression of *GmDREB2A;2* in soybean (*Glycine max*) resulted in improved performance in water-deficit experiments, higher rates in physiological parameters, and a trend of higher yield [243]. AT1G77450 is an upstream transcription factor of the MYB30 transcriptional cascade, which is a key regulator for *Arabidopsis* root-cell elongation. It also plays a role in stress-induced senescence under ROS signaling [244,245]. This transcription factor has also been identified in wheat, with a confirmed regulatory role in response to *Fusarium graminearum* infection [246].

A second example is the common stress target *AT3G14440* (*NCED3*), a gene in abscisic acid biosynthesis. The expression of *NCED3* was found to be significantly upregulated in TaHSFA2d-4A transgenic plants [247]. Wheat homologs of *Arabidopsis* NCED3 have also been associated with drought tolerance in wheat [248]. NCED3 has also been identified and its role in stress tolerance has been demonstrated in rice, maize, soybeans, and tomatoes [249,250,251,252,253].

The most represented group plays an important role in plant metabolism. Specifically, upregulated *AT3G14440* and *AT1G07430* have been associated with ABA biosynthesis and the regulation of ABA-signaling, supporting the role of ABA as a central regulator of abiotic stress resistance in plants [254]. A drought-tolerant wheat cultivar exhibited a smaller reduction in grain number and a smaller increase in endogenous ABA than a drought-sensitive cultivar [255]. Holesteens et al. (2022) confirmed the regulation of salt-stress responses by ABA in tomatoes [256]. Downregulated *AT4G29740* and upregulated *AT2G36750* play a role in the metabolism of cytokinins. This observation is consistent with the fact that heat, drought, salt, and cold stress impair the ability of plants to manage water [196]. A reduction in active cytokinin levels can be a result of cold and dehydration stresses in rice [257]. Positive effects of cytokinins on drought tolerance in wheat have been observed by Wang et al. [258] and are discussed, together with other hormones, in a later section.

AT5G11110 and AT3G13784 contribute to sugar synthesis. According to Pommerreging et al. (2018), sugar molecules can function as ROS scavengers and stabilizers of cell membranes and the osmotic cell potential [259]

The members of another strongly represented group play important roles in plant defense. For example, AT1G67360 is involved in the formation of *Arabidopsis* leaf-lipid droplets that are associated with secondary metabolite biosynthesis and plant resistance to stress [260]. Reduced expression of *AT5G03210* in *Arabidopsis* leads to increased susceptibility to the Plum pox virus [261]. AT2G32190 has been identified as the cysteine-rich transmembrane module that negatively regulates salt stress in *Arabidopsis* [262]. Downregulated *AT2G43550* encodes a defensin-like family protein, and a defense-like gene from winter wheat is possibly involved in enhanced tolerance against pathogens during cold acclimation [263]. The presence of genes related to the biotic stress response supports the notion of intersections between biotic and abiotic stress responses [264]. This phenomenon has been observed in *Arabidopsis*, rice, maize, wheat, and tomatoes [265,266,267,268,269].

In addition to the well-known genes involved in stress responses, our bioinformatic analysis also revealed five interesting candidates in stress responses without a characterized biological function: *AT1G67920*, *AT5G61820*, *AT5G15190*, *AT5G19875*, and *AT2G26530*.

## 4. Stress and Metabolites

Plants respond to abiotic stress by producing various metabolites that help them to deal with ever-changing environments [270]. Fine-tuning the production of primary metabolites, such as carbohydrates, organic and amino acids, is essential for plant growth, development, defense, and stress adaptation. Secondary metabolites (SM) are synthesized from the intermediates of primary carbon metabolism and are important for a plant’s defenses and its interactions with the environment. They include phenolics, carotenoids, flavonoids, sulfur-containing SM, nitrogen-containing SM, and volatile organic compounds [271]. In responding to stress, these compounds have roles as antioxidants, ROS scavengers, compatible solutes, and signal molecules, as well as in helping to maintain membrane stability. Plant hormones are small compounds derived from various essential metabolic pathways that regulate primary and secondary metabolism on the molecular scale. On the higher scales, phytohormones orchestrate growth, development, and interaction with the environment, including biotic and abiotic stresses [272]. Many studies of plant metabolome changes in response to abiotic stress have been conducted in recent years. Table 2 summarizes the metabolites involved in the responses to heat, drought, salinity, and cold, which have been identified in various crop species in the last six years (2017–2022). This review does not discuss ROS metabolism and signaling, which was comprehensively summarized in [217].

### 4.1. Primary Metabolites

#### 4.1.1. Amino Acids and Analogues

Changes in amino acid levels are associated with all abiotic stresses (Table 2). Of the amino acids, proline is a multi-functional molecule that accumulates in high concentrations in response to a variety of abiotic stresses and provides a supply of energy for plant growth once the stress is relieved [302]. Together with other osmolytes, it also mitigates the adverse effects of oxidative stress by scavenging oxygen species [303]. The use of transgenic plants engineered to synthesize proline more rapidly has been reported in numerous studies as having a positive influence on abiotic stress tolerance [304,305,306,307]. The main proline biosynthetic gene is PYRROLINE-5-CARBOXYLATE SYNTHASE (P5CS) and its stress-inducible expression has been shown to be a better strategy than constitutive expression. This is because negative effects, such as hampered growth and decreased productivity of transgenic plants, can be reduced [308]. Glycine-betaine, an amino acid analog, is another osmoprotectant that interacts with molecules involved in ROS scavenging and membrane stabilization and it also plays a role in the induction of stress-responsive genes [309,310]. Modification of the glycine–betaine biosynthetic pathway has also been shown to improve the abiotic stress resistance of crops. In plants, glycine–betaine is synthesized from choline and betaine aldehyde via oxidation mediated by the key enzymes choline monooxygenase and BETAINE ALDEHYDE DEHYDROGENASE (BADH). The overexpression of *BADH* is beneficial to drought tolerance in maize [311] and salt-stress tolerance in tomatoes [312] and it improves temperature-stress tolerance in wheat [313,314]. *BADH* genes are also present in species that do not accumulate glycine betaine such as *Arabidopsis* and rice. Two genes, *ALDH10A8* and *ALDH10A9*, have been identified to code for BADH in *Arabidopsis* and shown to function in the plant’s response to heat stress [315]. It has been suggested that *ALDH10A8* and *ALDH10A9* contribute to γ-aminobutyric acid (GABA) biosynthesis, which acts as a compatible solute and signal molecule and can alleviate the effects of various abiotic stresses [316,317].

#### 4.1.2. Organic Acids

Organic acids can regulate a broad range of cellular processes by modifying cellular pH and the cell redox state. They are also involved in the chemical modification of proteins, which alters their in vivo activity [318]. Besides their roles in development, nutrient uptake, and detoxification, their synthesis and metabolism are strongly influenced by abiotic stress conditions. Krebs cycle intermediates serve as direct markers of photosynthetic function and as mediators of osmotic adaptation. Salt-tolerant cultivars of broccoli showed higher concentrations of several Krebs-cycle metabolites, e.g., citric, succinic, malic, and fumaric acids, as well as the substrates of some anaplerotic reactions such as aspartic and glutamic acid [319]. Enhanced acetic acid accumulation in *Arabidopsis* improves its drought-stress tolerance via jasmonic acid signaling [320]. Endogenous citric acid levels increase in response to various types of abiotic stress [321,322] and the exogenous application of citric acid appears to alleviate the negative impact of these stresses on crop growth and yield [323,324].

#### 4.1.3. Carbohydrates

Sugars are not only a source of energy for the plant but they also act as regulators of various biological processes. In addition, they can serve as compatible solutes that protect cell membranes and proteins and maintain cell turgor pressure. Abiotic stress triggers the accumulation of soluble sugars and polyols in plants, including sucrose, glucose, trehalose, fructose, mannitol, sorbitol, and inositol [325].

Sucrose is the main form of assimilated carbon and the major transport molecule in higher plants. Under stress conditions, sucrose acts as an osmoregulatory molecule that prevents dehydration and regulates the expression of transcription factors and other genes involved in hormonal and defense signaling [326,327]. Since stress conditions may inhibit photosynthesis and thus limit the amount of sucrose supply, sucrose transporters (SUCs or SUTs) are a key component in securing sucrose distribution and plant stress tolerance [328,329].

Trehalose is a nonreducing disaccharide that can protect cell molecules by stabilizing biological membranes and proteins from environmental stress. TREHALOSE-6-PHOSPHATE SYNTHASE (TPS) and TREHALOSE-6-PHOSPHATE PHOSPHATASE (TPP) are two key enzymes that contribute to its biosynthesis in plants. Jiang et al. [330] generated OsTPP3-overexpressing rice plants that exhibited increased tolerance to simulated drought conditions as a result of changes in the expression of ABA biosynthetic and abiotic stress-related genes. Enhanced tolerance to heat stress has been observed in tomato plants overexpressing a trehalose-6-phosphate synthase/phosphatase fusion gene derived from *E. coli* [331]. An improvement in crop performance under osmotic stress by manipulating the levels of trehalose has been conferred in various other species such as common beans, maize, or soybeans [332,333,334]. Trehalose-6-phosphate (T6P), a precursor of trehalose, is an essential signal metabolite and acts as a regulator of sucrose levels in plants. Under changing environmental conditions, modification of T6P signaling seems to be an effective approach to boosting plants’ performance. The application of plant-permeable analogs of T6P to vegetative tissue improves recovery from drought [335].

Fructans are water-soluble polymers of fructose that are frequently correlated with improved freezing tolerance [336]. Their protective function is provided by their high water solubility and the resistance of fructan to membrane-damaging crystallization at freezing temperatures [337]. Starch represents the primary carbon reserve in plants. Dynamic changes in the starch–sugar interconversion enable plants to cope with abiotic stresses through the redistribution of energy and carbon when photosynthetic processes are limited [338]. Cellulose is the main component of the cell wall and its content has been shown to be reduced after exposure to salt stress [339]. Cell wall extensibility, which is provided by the relaxation of cell-wall polysaccharides, seems to be an important feature under stress conditions that enables cells to enlarge. The modification of cell-wall architecture is partially mediated by xyloglucan modification using XYLOGLUCAN ENDOTRANSGLUCOSYLASES/HYDROLASES (XTH, [340]). In tomato plants, the constitutive expression of a hot pepper (*Capsicum annuum*), xyloglucan endotransglucosylase/hydrolase *CaXTH3*, increased plant tolerance to salt and drought stresses without a negative impact on phenotype [341]. Pectins are acidic polysaccharides and their increased content in the cell-wall composition of root tips correlates with a higher salt tolerance of soybeans [342].

#### 4.1.4. Sugar Alcohols

Sugar alcohols, referred to as polyols, act both as osmotic regulators and redox balance maintainers that help plants deal with adverse conditions. One of them, mannitol, is present in many crop species but not all plants can synthesize it [343]. The exogenous application of mannitol to salt-stressed wheat has been shown to improve its salt tolerance by enhancing the activities of antioxidant enzymes [344], and the ectopic expression of the MANNITOL-1-PHOSPHATE DEHYDROGENASE gene (*mtlD)* for the biosynthesis of mannitol has been shown to improve wheat and peanut (*Arachis hypogaea*) tolerance to water stress and salinity [345,346]. Inositol is a cyclic polyol that has been shown to positively regulate cold tolerance in rapeseed by inhibiting CALCINEURIN B-LIKE1 (CBL1) and the induction of Ca^2+^ Influx [347]. The overexpression of the inositol biosynthetic gene *MYO-INOSTITOL-1-PHOSPHATE SYNTHASE* (*GsMIPS2*) from wild soybeans has been shown to increase the tolerance of *Arabidopsis* to salt stress [348]. Sorbitol is produced in parallel with sucrose during photosynthesis and serves as an energy translocation compound [349]. The exogenous application of sorbitol was shown to alleviate the negative effects of salt by reducing the H_2_O_2_ and malondialdehyde (MDA) contents in salt-sensitive tomatoes but had no positive effect on a salt-tolerant cultivar [350].

### 4.2. Secondary Metabolites

Secondary metabolites are derivatives of primary metabolites that are produced by plants. They can be divided into three major groups: phenolics, terpenes, and nitrogen/sulfur-containing compounds [351,352].

#### 4.2.1. Phenolics

Phenolics are ubiquitous aromatic compounds that have roles in plant-defense mechanisms against pathogens and abiotic stressors such as drought, salinity, and UV [353]. For example, in the total content of phenolic acids of wheat genotypes, flavonoids have been shown to increase as the growing temperature increases [354]. Under water-deficit stress, a stress-tolerant genotype of durum wheat has been shown to have higher total phenolic content in leaf tissue compared to stress-sensitive genotypes and higher total phenolic content in mature grains compared to a control [11]. In response to salt stress, total phenolic and flavonoid compounds have been shown to increase in wheat and maize [355,356]. The redirection of metabolic flux from lignin biosynthesis to flavonoid biosynthesis under salt, heat, and drought has been shown to lead to the accumulation of flavonoid glycosides in rice [232]. Thermal stress in tomato and watermelon plants has been shown to cause the accumulation of soluble phenolics [357]. The importance of flavonoids in freezing tolerance has been demonstrated in different *Arabidopsis* accessions [358].

#### 4.2.2. Terpenes

Terpenes perform complex roles in plant defenses against pathogens and herbivores [359], as well as other stressors. For example, terpenoids, specifically phytoalexins, accumulate in maize roots under drought and salinity stress, suggesting that they play a role in osmotic stress tolerance [360]; Bertamini et al. [361] observed a connection between monoterpene emission and heat-stress resistance in grapevines. Emissions of terpenes from tomato plants under salinity stress have been shown to increase in proportion to the salt concentration in the soil [362]. Mono- and sesquiterpene emissions have been shown to increase with the severity of cold and heat stress in tomatoes [363,364].

#### 4.2.3. Nitrogen/Sulfur-Containing Compounds

Nitrogen/sulfur-containing compounds include cyanogenic glycosides, alkaloids, and glucosinolates [351]. These compounds are known for their role in biotic stress resistance but they also play an important role in abiotic stress [365]. For example, drought stress in Chinese cabbage has been shown to induce the accumulation of glucosinolates in leaves, leading to stomatal closure [366], and glucosinolate metabolism has been shown to be overrepresented in wheat under the combined stresses of salt and heat [367]. Alkaloids are better known for their role in biotic stress resistance, but they also play important roles in oxidative stress [365,368] and abiotic stress since high temperatures during the initiation of flowering up to pod ripening have been shown to result in a higher alkaloid content of lupin seeds [369].

### 4.3. Phytohormones

Plant hormones are one of the most important elements in a plant’s ability to adapt to different environmental conditions. Although hormone molecules are produced in low concentrations, they are acutely sensitive to changing conditions and provide important short- and long-distance signals. Recently, stress-related phytohormones have been classified into nine groups—ABA, auxins (AUX), brassinosteroids (BR), cytokinins (CK), ethylene (ET), gibberellins (GA), jasmonates (JA), salicylic acid (SA), and strigolactones (SL) [370]. The biosynthetic pathways of phytohormones were recently comprehensively reviewed in [370]. Here, we focus mainly on their role in stress responses to abiotic stresses in crop plants.

Phytohormones can be classified according to many stress-related criteria. One of these is the significance of the phytohormone in plant stress resistance. Based on this criterion, hormones, together with their signaling cascades, can be divided into three groups: those that mainly increase resistance, those that decrease resistance to abiotic stresses, and hormones with highly context-dependent effects (Figure 8). ABA, SA, JA, BR, AUX, and SL can be classified as belonging to the first category. ABA is known to be a master resistance regulator for a range of stresses and plays an important role in water management. However, its signaling also modulates other aspects of stress resistance, such as the cold-responsive CBF regulon (Table 3). In addition to improved water management, plants under stress require the protection and maintenance of photosynthetic processes, control of the ROS level, and biosynthesis of protective compounds such as osmolytes, cryoprotectants, or scavengers of reactive species (Table 3). Another mechanism for increasing resistance to stress is the regulation of growth. AUX are hormones that have a significant impact on root growth and their asymmetric distribution plays an important role in responses to drought and salt stress (Table 3). This strategy may be slower than stomata closure but could be very important in longer time scales. It is not surprising that the phytohormones within this group show agonistic and synergistic properties with each other. Well-known examples are the positive effects of JA, SL, and BR on ABA [371,372,373] or interactions between BR and AUX [374]. Interestingly, melatonin has been shown to have a positive effect on stress resistance in various crops but it reportedly decreases the level of ABA, a process that is conserved across the different abiotic stresses [375].

GA is the only member of the second group. It is a hormone that has predominantly negative effects on plants under stress (Table 3). In non-stressful conditions, GA is an activator of plant growth. This function can be beneficial under some challenging conditions requiring growth such as thermomorphogenesis [376] but it seems that higher GA activity is not particularly compatible with stress resistance. Some hormones in the first group, for example, SA, also improve plant morphological traits (Table 3) but there is likely a difference between growth induced by reducing the negative impact of environmental cues and the direct induction of growth without effective stress reduction.

The effects of two hormone groups, cytokinins and ethylene, appear to be ambiguous. It seems that the effects of these hormones are highly dependent, not only on the plant species but also on the individual members of hormonal signaling, the plant tissue, and the specific conditions affecting the plant. It is, therefore, difficult to classify them into one of the previous categories; their action could be characterized as highly context-dependent. An ambiguous effect of ethylene can be seen, for example, in the response to cold. Ethylene has been shown to act as a negative regulator of freezing tolerance in *Arabidopsis* by repressing the expression of cold-responsive *CBF* genes [377]; however, elsewhere, it has been proven that applying 1-aminocyclopropane-1-carboxylate (ACC) can promote freezing resistance in grapevines through the upregulation of ethylene-responsive transcription factor *VaERF057* [378]. Cytokinins are an important class of hormones whose function is accompanied by intensive crosstalk with other hormones. The molecular mechanism of mutual antagonism between CK and ABA is well-described [379]. The inhibitory effect of CK on root growth suggests a negative role of CK on drought tolerance. This has been confirmed in some studies that utilized mutants of CK signaling in the model plant *Arabidopsis* [380,381]. However, under severe stress, CKs have also been shown to significantly increase drought resistance in tobacco and rice [382,383]. The effect of cytokinin is also dependent on its concentration because, although increased levels of CK activate the antioxidant system [384], a contrary effect has been observed in tobacco and *Arabidopsis* plants with high CK levels [385,386].

**Table 3 ijms-24-06603-t003:** Role of phytohormones in abiotic stress.

Hormone	Stress	Effect	Organism	Publication
ABA	heat	improved antioxidant system, lower MDA	wheat	[387]
	heat	higher yield	rice	[388]
	water stress	stomata closure, microtubules	thale cress	[120]
	osmotic stress	stomata closure	barley	[122]
	salinity	higher yield and water-use efficiency (WUE)	tomato	[389]
	cold	activation of CBF regulon	grapevine	[390]
	cold	improved antioxidant system	tomato	[391]
AUX	heat	increased yield	wheat	[392]
	heat	improved embryo development	rapeseed	[393]
	drought	decreased ROS, lower electrolyte leakage (EL)	soya	[394]
	osmotic stress	lower EL and MDA, increased chlorophyll	tobacco	[395]
	salinity	root growth	thale cress	[396]
	salinity	root growth	maize	[397]
	cold	increased proline, saccharides	rapeseed	[398]
BR	heat	improved growth, increased proline	wheat	[399]
	heat	improved antioxidant system	tomato	[400]
	drought	improved antioxidant system, ABA content	tomato	[373]
	osmotic stress	improved antioxidant system, ABA content	grapevine	[401]
	osmotic stress	higher survival, improved root growth	cotton	[402]
	salinity	higher WUE, increased proline	bean	[403]
	cold stress	photoprotection	tomato	[404]
	cold stress	improved antioxidant system, lower EL and MDA	tomato	[405]
CK	heat	higher yield	wheat	[406]
	heat	higher survival	thale cress	[407]
	heat	improved photosynthesis, higher proline	rice	[408]
	heat/drought	impaired photosynthesis, lower relative water content (RWC)	tomato	[409]
	drought	decreased survival, lower RWC	thale cress	[381]
	drought	higher yield	rice	[383]
	drought	improved antioxidant system	tobacco	[382]
	salinity/drought	decreased survival	thale cress	[410]
	salinity	improved photosynthesis, lower MDA	tomato	[411]
	salinity	improved photosynthesis and growth, lower EL	rice	[412]
	cold stress	induction of cold-responsive genes	maize	[413]
	cold stress	increased and also decreased survival	thale cress	[414]
ET	heat	lower membrane oxidation and EL, higher biomass	rice	[415]
	heat	higher pollen quality	tomato	[416]
	salinity	increased ROS, inhibited root growth	rice	[417]
	salinity	increased ROS	tobacco	[418]
	salinity	increased sensitivity to stress	cucurbits	[419]
	salinity	improved Na/K homeostasis	thale cress	[420]
	drought	drought-induced senescence	maize	[421]
	drought	increased survival	rice	[422]
	drought	lower yield	barley	[423]
	drought	lower yield	maize	[424]
	cold stress	increased survival	grapevine	[378]
	cold stress	repressed CBF	thale cress	[377]
GA	heat	positive role in thermomorphogenesis	thale cress	[376]
	heat	higher EL, impaired photosynthesis	barley	[425]
	drought	decreased RWC	tomato	[426]
	drought	lower yield and pigments	cereals	[427]
	salinity	root differentiation/decreased tolerance	thale cress	[428]
	cold	increased EL, impaired antioxidant system	maize	[429]
	cold	decreased CBF expression	thale cress	[430]
	cold	decreased EL and MDA, mitigated stress	tomato	[431]
JA	heat	improved photosynthesis	wheat	[432]
	heat	increased survival, improved photosynthesis	thale cress	[433]
	drought	increased biomass, higher water content	tomato	[434]
	drought	higher antioxidant system, increased proline	sweet potato	[435]
	salinity	decreased Na^+^ concentration	barley	[436]
	salinity	increased proline, higher tolerance	sorghum	[437]
	cold	increased ABA, lower EL, improved photosynthesis	tomato	[371]
	cold	increased sugars, decreased browning index	peach fruit	[438]
SA	heat	improved antioxidant system, lower MDA	wheat	[387]
	heat	protected from pollen abortion, decreased ROS	rice	[439]
	drought	lower EL and MDA, higher RWC	barley	[440]
	drought	increased yield	tomato	[441]
	salinity	improved antioxidant system, lower Na+ level	potato	[442]
	salinity	increased yield	tomato	[443]
	cold	improved photosynthesis, lower EL and ROS	wheat	[444]
	cold	lower EL, improved antioxidant system	grapevine	[445]
SL	heat/cold	higher ABA content, increased resistance	tomato	[372]
	heat	higher germination, higher proline level, lower MDA	lupine	[446]
	drought	improved growth, higher chlorophyll, higher RWC	barley	[447]
	drought	improved photosynthesis, lower ROS	wheat	[448]
	salinity	improved antioxidant system and growth	tomato	[449]
	salinity	improved antioxidant system and photosynthesis	cucumber	[450]
	cold	lower ROS and MDA, increased proline	mung bean	[451]
	cold	improved antioxidant system and photosynthesis	rapeseed	[452]

### 4.4. Other Growth Regulators

Several other growth regulators are known, which are gradually coming to the attention of scientists. Among them, polyamines and melatonin have been studied intensively.

Melatonin has been known for almost 20 years; however, its potential role as a phytohormone with multiple physiological actions has only recently emerged [453]. Interestingly, melatonin has been shown to have a positive effect on stress resistance in various crops [454,455,456] but it reportedly decreases the level of ABA, a process that is conserved across the different abiotic stresses [456,457].

Polyamines, such as putrescine, spermidine, and spermine, are small organic molecules that are typically elevated in plants under abiotic stress conditions [458,459,460,461]. Moreover, it has been shown that their exogenous application could increase tolerance to drought or cold stress [103,462].

Many studies have shown that phytohormones and other growth regulators can improve plant performance under abiotic stress. Thus, the modulation of their metabolism by inhibitors and activators is a promising strategy for protecting plants from yield losses.

## 5. Conclusions and Future Prospects

Research in recent decades has successfully described some of the molecular mechanisms underlying plant resistance to different abiotic stresses. In this review, we focused on the novel mechanisms of plant resistance to four fundamental environmental factors—drought, heat, cold, and salinity. The majority of the reviewed mechanisms, including stress signaling, hormonal regulation, and metabolic changes, have been revealed in the model plant *Arabidopsis* but not all mechanisms can be easily applied to crop plants. For example, differences can be found in the effects that plant hormones have on different plant species under stress conditions (Table 3). Novel and species-specific defense mechanisms could be revealed through modern statistical analyses such as genome-wide association studies [463] and further progress can be expected in the identification, characterization, and confirmation of promising targets in the near future. In addition to the analysis of different alleles or single nucleotide polymorphisms, differences in the stress tolerance of younger and older tissues within the same plant body suggest that gene and protein regulation may still teach us many things about plant resistance.

The specificity of calcium signaling remains an open question. It has been shown that various stimuli can activate diverse calcium signatures, resulting in specific gene regulations. However, parameters such as the period or amplitude of the calcium signature cannot fully explain the molecular mechanism of the selection of targets to be activated. In field conditions, plants are usually exposed to multiple stresses. Similar to previous works, our meta-analysis has shown that targets of gene expression can be regulated differently under the influence of different stressors (Figure 4). Thus, experiments employing multiple stress conditions and their performance under field conditions are expected to confirm known mechanisms or raise new questions concerning plant stress resistance in nature. An important factor in stress resistance is light. Light is, of course, the primary source of energy for plants but stress responses are also significantly modulated by light conditions [464]. However, laboratory experiments typically use well-controlled and homogeneous light conditions that differ significantly from the natural variations in the field. As a result, the effects of light on plants in the field are not well-understood. Another aspect of field conditions that could have both stimulatory and inhibitory effects on plants and their response to the environment is the presence of a potent microbiome. Recent research has confirmed the importance of bacteria in plant tolerance [270]. The interactions between plants and other organisms, including hormonal regulations, are very complex and likely highly species-specific but continuing to advance our understanding of them is highly beneficial as we strive to improve the sustainability of our agriculture systems.

The presented meta-analysis also showed that although the number of genes regulated similarly in response to all cardinal stress factors is small, some of them have not yet been characterized. However, whether any of these genes are as critical as DREB2A or NCED3 is a topic for future research.

## Figures and Tables

**Figure 1 ijms-24-06603-f001:**
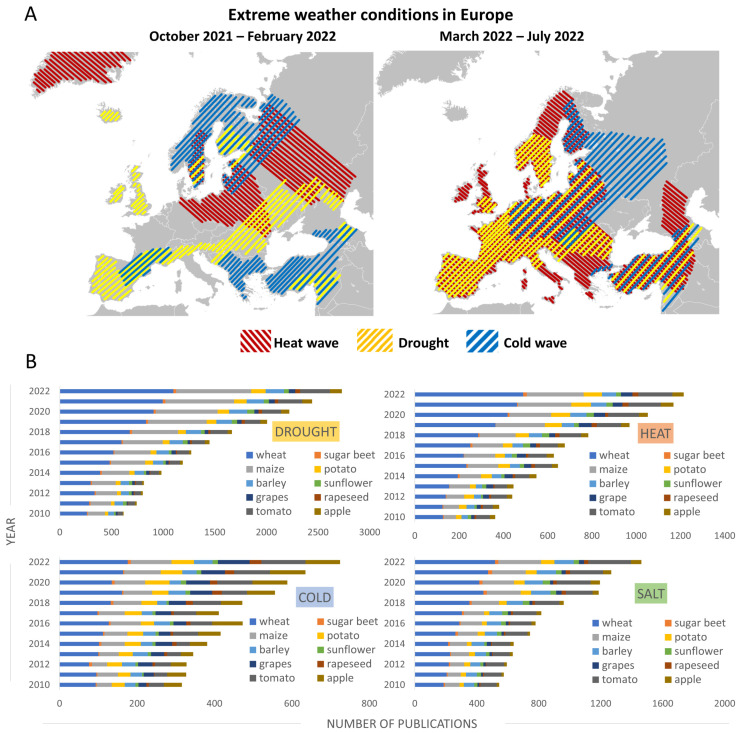
Locations of recorded extreme weather conditions in Europe and the number of published works focusing on abiotic stress in crops. (**A**) Weather conditions recorded by the German Weather Service from October 2021 to February 2022 and March 2022 to July 2022 (Deutscher Wetterdienst, 2022, [15]). The heat or cold wave is a period of at least one week with temperature anomalies (exceeding +6 °C over or −6 °C below the average temperature during the reference period 1981–2010). Panel 1A was adapted from an original map image made by Maix, Wikimedia Commons, distributed under a CC SA 3.0 License. (**B**) The number of publications focusing on a specific crop and abiotic stress in the last 13 years. The data were obtained from the Clarivate Web of Science database based on the methodology detailed in Appendix A.

**Figure 2 ijms-24-06603-f002:**
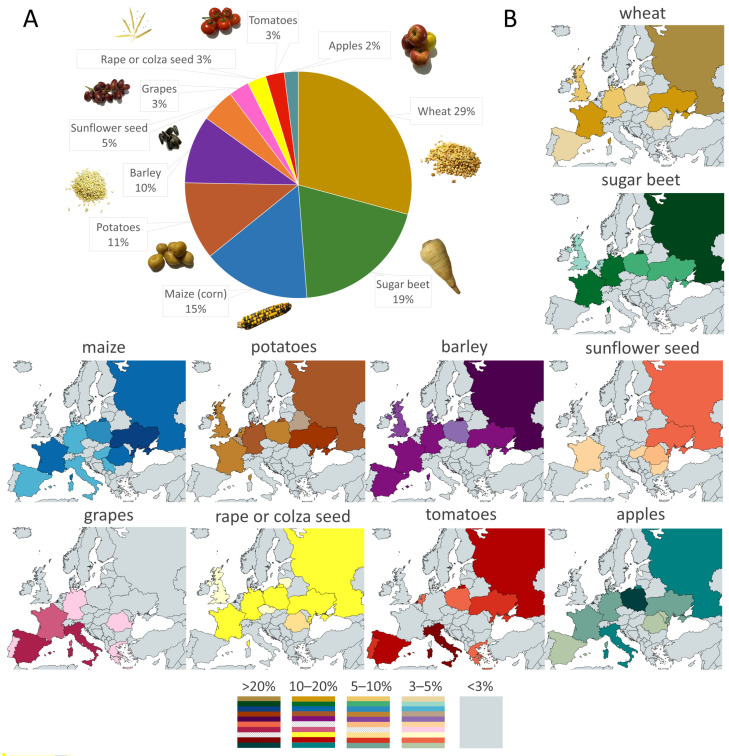
The most important crops produced in Europe in 2021. Data were obtained from FAOSTAT [8]. (**A**) The pie chart represents the percentage of various crops under production in Europe in 2021. (**B**) The main countries in Europe that produce the selected crops. The color intensity corresponds to the contribution of individual states to overall production in Europe. The color coding in the legend was sorted from top to bottom according to the amount of production of individual crops. Hatched rectangles in the color legend indicate crop production not used in the figure. Panel 2B was prepared using maps from www.mapchart.net (accessed on 17 January 2023), distributed under a CC SA 4.0 License.

**Figure 3 ijms-24-06603-f003:**
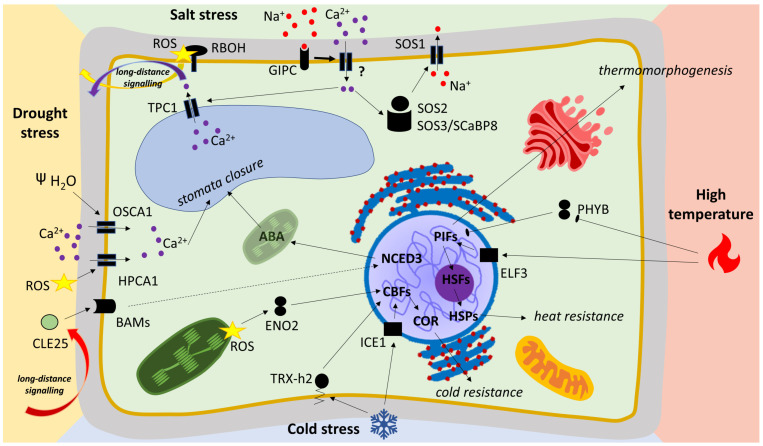
Abiotic stress signaling. Image represents novel mechanisms in abiotic stress signaling reviewed in corresponding sections.

**Figure 4 ijms-24-06603-f004:**
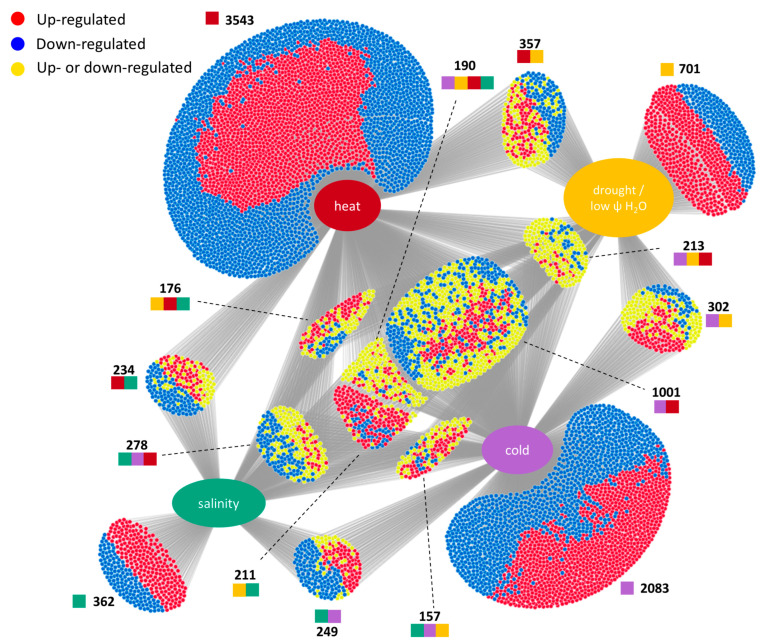
Visualization of groups of genes found in response to abiotic stressors. Data based on *Arabidopsis* expression profiles found in the Expression Atlas database [203]. Since plants are sensitive to stress with respect to the developmental stage, our analysis included only those works that met the following criteria: (1) involved the model plant *Arabidopsis thaliana*, (2) seedling stage, and (3) application of one stress. In total, the database included 11 transcriptional studies for further analyses (2 studies of heat-responsive transcriptome [204,205], 3 studies of cold-responsive transcriptome [206,207,208], 3 studies of drought-responsive transcriptome [209,210,211], and 3 studies of salt-responsive transcriptome [204,212,213]). Details of the analysis are summarized in Appendix A. Circles represent genes (red, upregulated; blue, downregulated; yellow, inverse response for different stresses). Squares represent stress treatments. Genes responsive to single abiotic stress are marked by one square, with the color corresponding to the stress (red, heat; orange, drought/low water potential; green, salinity; violet, cold). Genes responsive to several stressors are visualized by clusters marked with corresponding squares that represent all stress modulators. The image was prepared in DiVenn 1.2 [214].

**Figure 6 ijms-24-06603-f006:**
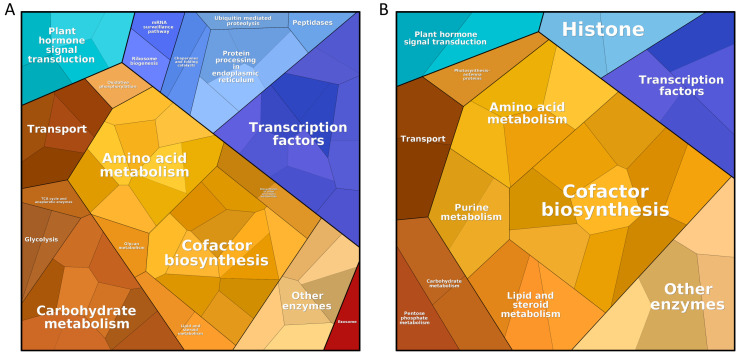
Functional groups of genes with similar stress regulation in at least three abiotic stresses (identified in Figure 4): (**A**) 222 genes upregulated in at least 3 stresses; (**B**) 197 genes downregulated in at least 3 stresses. Visualization was prepared using proteomaps [222].

**Figure 7 ijms-24-06603-f007:**
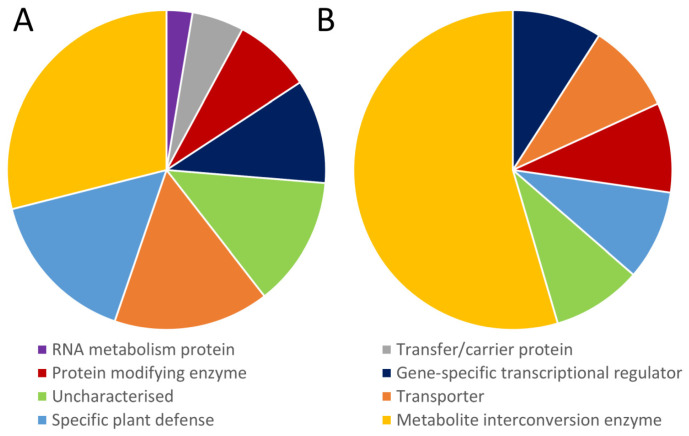
Functional annotations of common genes with changed regulation during stress crosstalk (heat, cold, drought, salinity; identified in Figure 4): (**A**) Protein classes represented in 38 common genes upregulated during stress crosstalk. (**B**) Protein classes represented in 11 common genes downregulated during stress crosstalk [240].

**Figure 8 ijms-24-06603-f008:**
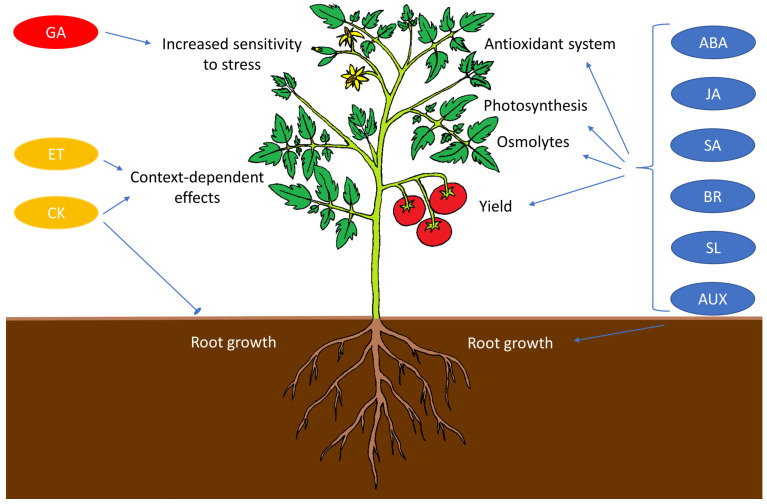
Role of phytohormones in plant stress response. Hormones with an overall positive effect are visualized by blue ovals. Hormones visualized by yellow ovals have been shown to have positive and negative effects on plant stress resistance in different studies. GAs visualized by red ovals mainly increase sensitivity to abiotic stress. More details about the function of hormones in stress responses in crops are shown in Table 3.

**Table 1 ijms-24-06603-t001:** Optimal temperature conditions for the principal European crops.

Crop	Optimal Temperature	Developmental Stage	Publication
Wheat (*Triticum aestivum*)	12–22 °C	flowering and grain filling	[37]
Barley (*Hordeum vulgare*)	25 °C	grain filling	[38]
Maize (*Zea mays*)	28–32 °C	general	[39]
Rapeseed (*Brassica napus*)	21–25 °C	general	[40]
Sugar beet (*Beta vulgaris*)	22–26 °C	growth of the taproot	[41]
Potatoes (*Solanum tuberosum*)	15–19 °C	tuber growth	[42]
Grapes (*Vitis vinifera*)	20–40 °C	berry development	[43]
Tomatoes (*Solanum lycopersicum*)	22–26 °C	fruit set and satisfactory fruit yield	[44]
Sunflowers (*Helianthus annuum*)	12–20 °C	general	[45]
Apples (*Malus domestica*)	18–21 °C	shoot growth and floral initiation	[46]

**Table 2 ijms-24-06603-t002:** Changes in metabolite levels and their correlations with stress resistance. Metabolites involved in responses to heat, drought, salinity, and cold stress in selected crop species identified in the last six years (2017–2022). The changes in the metabolites after exposure to stress are visualized by arrows. Upward-arrow (↑) means increase and downward-arrow (↓) means decrease in the metabolite level.

Crop	Description of the Experiment	Metabolites Involved in the Response to Stress Conditions
heat
potato (*Solanum**tuberosum*)	Metabolome changes after 3 days of heat stress (35 °C) in potato leaves [273].	↑ tyrosine, arachidonic acid metabolism, flavone, and flavonol biosynthesis
↓ glutathione, linoleic acid, steroid, fatty acid, phosphonate, and phosphinate metabolism
maize(*Zea mays*)	Metabolome change after long-term heat stress in maize leaves. Heat stress started at 30/24 °C, which was increased 2 °C per day for 5 days, then maintained at 37 °C for 12 days [274].	↑ tryptophan, threonine, histidine, raffinose, galactinol, lactitol
↓ citrate and trans-3-caffeoyl quinic acid
wheat(*Triticum**aestivum*)	Two contrasting spring wheat genotypes were exposed to heat stress (34/16 °C, 10 days) during heading. Anthers were collected for metabolic profiling [275].	↑ N-based amino acids, ABA, IAA-conjugate
↓ dehydroascorbic acid, quinic acid, 5-Hydroxyindole-3-acetic acid, putrescine, and shikimic acid
grape(*Vitis vinifera*)	Metabolomic analysis of high-temperature effect (34/26 °C, 14 days) on grapevine berries [276].	↑ lipid metabolism metabolites, lignin, cuticle, vax, GABA, galactinol, vitamins
↓ malic acid, shikimate, sugar phosphate, secondary metabolites, sugars
maize(*Zea mays*)	Recovery profiling following sudden heat shock (46 °C, 2 h) regarding metabolites in two maize genotypes grown under ambient or elevated CO_2_ [277].	↑ ribose, valine, asparagine, isoleucine, adipic, 2-oxoglutarate, pyruvate, maltose, malate, trehalose, myo-inositol, starch, citric, fumarate
↓ glycerate, serine, glycine, shikimate, leucine, proline, and sucrose
tomato(*Solanum* *lycopersicum*)	Untargeted metabolomic analyses of tomato pollen after short heat exposure (38 °C, 2 h) [278].	↑ flavonoids

drought
sunflower(*Helianthus* *annuus*)	Comparison of metabolic profiles of sensitive/tolerant sunflower seedlings subjected to water-deficit stress [279].	Water-deficit stress-tolerant line accumulated:
↑ anthranilic acid, maleic acid, malonic acid, putative-rhamnose, fructose
wheat(*Triticum**aestivum*)	Drought effect (up to 10 days after withholding water) on bread wheat metabolism during the flowering stage [280].	↑ 1-aminocyclopropane-1-carboxylic acid, Asn, serotonin, GABA, cystine, deoxyuridine, tryptamine, putrescine
↓ glyceric, shikimic, ferulic and succinic acid
barley(*Hordeum**vulgare*)	Transcriptome and metabolome analysis on the developing grains of two barley genotypes differing in the responses to drought stress [281].	↑ amino acids, sugars, abscisic acid, jasmonic acid, ferulate
↓ citrate
wheat(*Triticum**aestivum*)	Metabolic adjustment of six winter wheat cultivars to drought (induced by withholding watering for 6 days) [282].	↑ sugars, malic acid, oxalic acids, proline, threonine, GABA, glutamine, myo-inositol
↓ propanoic acid
wheat(*Triticum**aestivum*)	Changes in protein and metabolite abundance of two wheat cultivars after 7 days of water deficit [230].	↑ purine bases, organic acids, sugars, amino acids
↓ aspartate, glutamate
barley(*Hordeum* *vulgare*)	Metabolic changes in four wild and cultivated barley genotypes contrasting in drought tolerance during grain-filling stage in response to water stress [283].	↑ mannitol, L-proline, sucrose, TCA cycle components, quinic acid
↓ 2-ketoglutaric acid
potato(*Solanum**tuberosum*)	Set of predictive markers for drought tolerance by transcriptomic and metabolomic profiling of 31 potato cultivars [284].	markers for drought tolerance:
↑ galactaric, galactonic, glyceric, saccharic acid, dopamine, tyramine
sunflower(*Helianthus* *annuus*)	Metabolic pathways related to drought conditions in sunflowers. The response of plants was studied in the early stage of water deficit [285].	↑ TCA cycle components, carbohydrates, amino acids, and derivatives proline, tyramine, glycine, malonate, γ-aminobutyrate
↓ amino acid metabolites
salinity
barley(*Hordeum* *vulgare*)	Metabolic analyses of barley seeds in response to salt stress (24 h, 200 mM NaCl), during the germination process. Two differentially salt-tolerant barley varieties were compared [286].	↑ aminoacyl-tRNA biosynthesis, glycine, serine and threonine metabolism, glyoxylate and dicarboxylate metabolism, and porphyrin and chlorophyll metabolism (tolerant)
↑ valine, leucine and isoleucine biosynthesis, biosynthesis of amino acids, alanine, aspartate and glutamate metabolism, glycine, serine and threonine metabolism, and cyanoamino acid metabolism (sensitive)
rapeseed(*Brassica napus*)	Molecular mechanism of salt tolerance in rapeseed. Two rapeseed varieties were compared, showing the metabolites common to both [287].	↑ glutathione amid, aconitase, glucose, mannose, inositol, epigallocatechin 3-gallate
↓ arginine, citrulline, trimethyl-lysine, acetylaspartate, inositol-triphosphate
rapeseed(*Brassica napus*)	Key salt-related metabolites in five different rapeseed cultivars. Salt stress (up to 200 mM NaCl) was applied during the early seedling stage [288].	↑ linolenic acid, xanthosine, inosine 5′-monophosphate, adenosine 3′-monophosphate, niacinamide, oleamide, phosphoric acid, etamiphylline (in tolerant cultivars)
↓ 5-hydroxytryptophan, cholesterol, L-aspartic acid, beta-homotreonine, N-p-coumaroyl serotonin, ornithine (in tolerant cultivars)
sugar beet(*Beta vulgaris*)	Metabolites involved in the short-term (1 day) and long-term (7 days) salt-stress response (300 mM Na+ treatment) in sugar beet [289].	↑ L-malic acid and 2-oxoglutaric acid, amino acids, betaine, melatonin, (S)-2-aminobutyric acid, cis-aconitate, benzoic acid L-malic acid, alpha-ketoglutarate, 2-isopropylmalic acid
↓ sucrose
barley(*Hordeum* *vulgare*)	Ionomic, metabolomic, and proteomic responses in roots of salt-tolerant/sensitive barley accession exposed to salinity stress (200 and 400 mM) [290].	↑ fructose, trehalose, sorbitol (in both genotypes), glycine, alanine, valine, inositol, allothreonine, glutamic acid, glycine, cysteine (in tolerant genotypes)
↓ glucose-6-P, fructose-6-P
durum wheat(*Triticum* *durum*)	Metabolic changes in the shoots and roots of five durum-wheat genotypes exposed to the different salt levels [291].	↑ proline
↓ organic acids involved in the Krebs cycle, gluconic, quinic, shikimic acid
sugar beet(*Beta vulgaris*)	Metabolic adaptation of sugar beet to salt stress (up to 300 mM NaCl) at the cellular and subcellular levels. Metabolites were profiled at 3 h and 14 d after reaching the maximum salinity stress [292].	↑ arabinose, gluconolactone, inositol, mannitol, proline, serine, and thymine
↓ lactate, homoserine, adenosine, guanine (early response), fumarate, L-aspartate, gluconate (late response)
barley(*Hordeum* *vulgare*)	The effects of salinity stress (up to 150 mM NaCl) on barley roots through quantitation of polar metabolites [293].	↑ 4-hydroxy-proline, alanine, arginine, asparagine, citrulline, glutamine, phenylalanine, proline
↓ putrescine, succinate
cold
maize(*Zea mays*)	Metabolic responses under cold stress (5 °C, 24 h) in the early growth stages of maize. Responses of tolerant and susceptible lines were compared [294].	Cold-tolerant line accumulated:
↑ guanosine 3′,5′-cyclic monophosphate, quercetin-3-O-(2″′-p-coumaroyl)sophoroside-7-O-glucoside, phloretin, phloretin-2′-O-glucoside, naringenin-7-O-Rutinoside, L-lysine, L-phenylalanine, L-glutamine, sinapyl alcohol, feruloyl tartaric
apple(*Malus**domestica*)	Molecular mechanism of apple trees in response to freezing injury during winter dormancy. Cold-resistant and cold-sensitive cultivars were compared [295].	↑ 4-aminobutyric acid, spermidine, and ascorbic acid (cold-resistant)
↓ oxidized glutathione, vitamin C, glutathione, spermidine (cold-resistant)
rapeseed(*Brassica napus*)	Metabolite profiling of cold-treated (−2 °C, 2 h) contrasting rapeseed genotypes focusing on siliques [296].	↑ 8-hydroxyguanosine, 9-(arabinosyl)hypoxanthine, inosine, uridine, guanosine 3′,5′-cyclic monophosphate, β-pseudouridine, 4-acetamidobutyric acid, phenylpyruvic acid, 6-hydroxyhexanoic acid, valeric acid, γ-aminobutyric acid, oxalic acid, jasmonic acid (both genotypes)
↑ adenine, riboprine, cytidine, N6-isopentenyladenine (cold-tolerant only)
maize(*Zea mays*)	Metabolic responses of maize hybrids could be extrapolated from growth-chamber (gradually decreasing temperature) conditions to early sowing in the field [297].	↑ trans-aconitate, coumaroyl hydroxycitrate, chrysoeriol glucosyl rhamnoside, caffeoylquinate, ferruloylquinate, (iso)vitexin, DIBOA-glucoside
↓ malate, glutamine
rapeseed(*Brassica napus*)	Cold-responsive metabolites in two contrasting varieties of rapeseed after 1 and 7 days of cold treatment [298].	Response to cold in both varieties:
↑ trehalose, L-Kynurenine, gamma-tocotrienol, phenyllactic acid, L-Gulonic gamma-lactone
maize(*Zea mays*)	Two maize lines with contrasting chilling-tolerance capacities were used to identify the major factors of chilling tolerance. The plants were exposed to 14 °C day/10 °C night for 60 days [299].	Chilling tolerance in tolerant plants correlated with:
↑ chlorophyll content, glucose-6-phosphate dehydrogenase activity, sucrose-to-starch ratio
wheat(*Triticum**aestivum*)	Metabolite activity in winter-hardy wheat subjected to cold stress (cold acclimation at 4 °C for 28 days, then freezing at −5 °C for 24 h) [300].	↑ aspartic acid O-rutinoside, proline, tyramine, raffinose, gluconic acid, melezitose, mannose, maltotetraose
↓ aspartic acid, lysine, ornithine
potato(*Solanum**tuberosum*)	Metabolome of the freezing-tolerant *Solanum acaule* and freezing-sensitive *S. tuberosum*. The plants were exposed to 4 °C for 14 days, then to gradient freezing at 1 °C/h up to −12 °C [301].	Chilling tolerance in tolerant plants correlated with:
↑ putrescine, 1-kestose, raffinose, xylose, fucose, isoleucine, tyrosine, valine, benzoic acid, trans-caffeic acid, dehydroascorbic acid, uracil

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
