# Peer review of "Abiotic Stress in Crop Production"

_ijms, 2023, doi:10.3390/ijms24076603_

Round 1

Reviewer 1 Report

The review is very interesting and actual focusing on the problem of crop production in relation to climate change. Nevertheless, a revision is necessary before its publication.

- All the publications (complete bibliographic citation) used for figure 1b must be reported as additional material. The publications must be subdivided by stress per crop as shown in fig 1b. Having made the figure 1b, the authors should easily retrieve the bibliographic information.

- Meta-analysis: it seems that the authors have made a meta-analysis, and as result a diagram of venn (Fig 4) is reported. Nevertheless, more details on metanalysis procedure must be reported to statistically validate the results successively represented in the venn diagram (fig 4). For this see some papers describing meta-analysis procedure and validation (Moher, et al 2009. Phys. Ther., 89, 873–880. https://doi.org/10.1093/ptj/89.9.873;  Nakagawa, et al 2017. BMC Biology, 15(1), 18. 10.1186/s12915-017-0357-7; Rosenberg M, Adams D, Gurevitch J (2000) MetaWin: statistical software for meta–analysis cited in Yang et al 2023. Plant Biotechnology Reports (2023) 17:39–52, https://doi.org/10.1007/s11816-022-00770-0 ).

Moreover, considering that this is a review, important meta-analysis applications can be found in google scholar using the key words “meta-analyses, abiotic stress response” and asking only articles published from 2022 to present. Please, see also all these articles with the application of meta-analysis on abiotic stress response in plants, and cite them in bibliography.

A complete description of meta-analysis procedure and related statistical analysis must be reported as additional material.

Moreover, the title of Fig 4 must better explain the figure and all the connections among stresses etc. Intuitively, the direct connection with a group of genes and a stress, should be unique genes related to that stress (example 362 genes unique genes for salinity), and group of genes are in common with two (example 234 genes between salinity and heat) or more stress (example 176 genes among salinity, heat, and drought?), etc.

It seems a very high number of unique genes for each stress and may be this could be because a statistical meta-analysis has not been applied and therefore there is an overestimation of unique genes.

- The number of genes reported in figure 4 are not the same reported in the text from lane 490 to lane 501: for example, cold regulated 4473 but in the figure the total number for cold seems to be 2083, etc. also for the other stresses there is not a correspondence between the figure and the text. These must be checked, but first a statistical analysis must be done before to realize the figure 4.

Author Response

We would like to thank all reviewers for the positive evaluation of our work; we appreciate it very much. We would also like to thank you for the comments and suggestions that significantly improved the manuscript. We believe that the edited version includes all the required information.

Response to Reviewer 1

1.1 - All the publications (complete bibliographic citation) used for figure 1b must be reported as additional material. The publications must be subdivided by stress per crop as shown in fig 1b. Having made figure 1b, the authors should easily retrieve the bibliographic information.

With permission from Clarivate WOS, we modified Figure 1B to focus only on data from the WOS collection, from which individual articles can be easily generated. The exact settings for the individual filters used are part of supplemental material 1. Figure 1 was replaced by a modified version.

 1.2a - Meta-analysis: it seems that the authors have made a meta-analysis, and as result a diagram of venn (Fig 4) is reported. Nevertheless, more details on metanalysis procedure must be reported to statistically validate the results successively represented in the venn diagram (fig 4). For this see some papers describing meta-analysis procedure and validation (Moher, et al 2009. Phys. Ther., 89, 873–880. https://doi.org/10.1093/ptj/89.9.873;  Nakagawa, et al 2017. BMC Biology, 15(1), 18. 10.1186/s12915-017-0357-7; Rosenberg M, Adams D, Gurevitch J (2000) MetaWin: statistical software for meta–analysis cited in Yang et al 2023. Plant Biotechnology Reports (2023) 17:39–52, https://doi.org/10.1007/s11816-022-00770-0 ).

and

1.2b A complete description of meta-analysis procedure and related statistical analysis must be reported as additional material.

We agree that we should provide more accurate information about the methodology. Readers can find a description, including a graphic overview, in the supplementary material 2 file and also in the cited reference.

1.3 Moreover, considering that this is a review, important meta-analysis applications can be found in google scholar using the key words “meta-analyses, abiotic stress response” and asking only articles published from 2022 to present. Please, see also all these articles with the application of meta-analysis on abiotic stress response in plants, and cite them in bibliography.

We agree that we could include other meta-analyses related to the topic. Some of them have been included in the main body of the text.

1.4 Moreover, the title of Fig 4 must better explain the figure and all the connections among stresses etc.

and

1.5a Intuitively, the direct connection with a group of genes and a stress, should be unique genes related to that stress (example 362 genes unique genes for salinity), and group of genes are in common with two (example 234 genes between salinity and heat) or more stress (example 176 genes among salinity, heat, and drought?), etc.

and

1.5b - The number of genes reported in figure 4 are not the same reported in the text from lane 490 to lane 501: for example, cold regulated 4473 but in the figure the total number for cold seems to be 2083, etc. also for the other stresses there is not a correspondence between the figure and the text. These must be checked, but first a statistical analysis must be done before to realize the figure 4.

We have now described the figure legend in greater detail so that the results are properly understood. Number 4473 represents all cold-responsive genes and a fraction of 2083 genes is uniquely responsive to the cold. Additional information not visible in Figure 4 is mentioned in the text, the text is a complement of Figure 4. We believe that legend is more informative now.

1.6 It seems a very high number of unique genes for each stress and may be this could be because a statistical meta-analysis has not been applied and therefore there is an overestimation of unique genes.

We believe that our analysis does not violate common standards. Compared to other works such as 10.3390/plants11040502 we have used less strict conditions for significant changes (p-value 0.05 vs. 0.01), and this can result in the identification of a higher number of the DEGs and also unique genes. But still, the number of genes responsive to only one stress condition is not surprising if we consider the currently published results for wheat (10.1007/s00709-022-01807-5).

Reviewer 2 Report

In the current study, the authors provided a detailed insight of the abiotic stress in crop production its effects on the production and yield, its management, and mechanisms. In addition, the authors also explored the pathways of the stress. The study topic is very interesting, and provide many gaps for the future studies. The overall manuscript is well organized and well-written. However, the main concerns about this manuscript can be found below, and minor revision are suggested.

In abstract main findings should be written and how the study was carried out.

Specify the stresses which have been focused in this review.

Line 23 “major crop plants” write names of the crops.

In first second and third paragraph of the introduction the author discussed the climate change, it should be link to stress factors and its impacts on crop production.

Line 66 should be cited with relevant study “https://doi.org/10.1016/j.indcrop.2022.116090

The authors are recommended to include a heading at each section of alternative strategies or management strategies against respective stress.

Line 506 to 510 should be cited with recent study https://doi.org/10.3390/antiox12020268

Conclusion of the study is missing.

Also add missing links and gaps in future prospects.

Author Response

We would like to thank all reviewers for the positive evaluation of our work; we appreciate it very much. We would also like to thank you for the comments and suggestions that significantly improved the manuscript. We believe that the edited version includes all the required information.

Response to Reviewer 2:

2.1a In abstract main findings should be written and how the study was carried out.

and

2.1b Line 23 “major crop plants” write names of the crops.

Since other reviewers did not request changes to the abstract, we made only minor changes to be more accurate and informative.

2.3 Specify the stresses which have been focused in this review.

Stresses are specified on line 12

2.4 In first second and third paragraph of the introduction the author discussed the climate change, it should be link to stress factors and its impacts on crop production.

We thank the reviewer for this suggestion which is similar to point 3.2 (reviewer 3). We have modified this part to make it more complex, as suggested.

2.5 Line 66 should be cited with relevant study “https://doi.org/10.1016/j.indcrop.2022.116090

This relevant recent study was used, as recommended.

2.6 The authors are recommended to include a heading at each section of alternative strategies or management strategies against respective stress.

We have added the following headings:

2.2.4. The role of root growth in drought resistance

2.2.5. Roots and hydrotropism

2.2.6. Long-distance signaling of water deficit

2.2.7. Other mechanisms of drought resistance

2.4.3. Role of redox changes in cold signaling

2.7 Line 506 to 510 should be cited with recent study https://doi.org/10.3390/antiox12020268

We are grateful for the recommendation of the very recent reference.

2.8a Conclusion of the study is missing.

and

2.8b Also add missing links and gaps in future prospects.

Motivated by several recent reviews (10.3390/plants11212919; 10.1093/jxb/erab549; 10.3390/w15040739), we have finished the review by combining chapter Conclusions and Future prospects, and we modified it accordingly.

Reviewer 3 Report

In the current climate change era, abiotic stresses significantly impact crop production worldwide. Thus, it is vital to understand stress responses and tolerance mechanisms. Overall, the authors have done a great job and prepared a very attractive review. I have some suggestions for further improvement.

Throughout the text, please replace resistance/resistant with tolerance/tolerant. Resistance/resistant usually refers to biotic stresses.

In the introduction, authors can provide a little information about how a single or the combination of these abiotic stresses impacts agricultural output and food safety, most probably in the first paragraph.

In Table 1/2, and the main text and other parts, please also mention the botanical names within the parenthesis on the first mention.

The number of references is too high. Authors can replace some older references with new ones, and the same citation of some review articles can be repeated at different places instead of using the new reference(s) for basic statements. Here, I suggest some of the recently published articles regarding different abiotic stresses that need to be adjusted to provide up-to-date references such as doi: 10.1111/plb.13510;  10.1093/jxb/erac073; 10.1002/tpg2.20279; 10.1111/tpj.15483; etc. Authors can also check some high-impact papers and please try to reduce some basic references with the above suggestions and use them multiple times.

Please carefully check the whole text and define the undefined abbreviations on the first mention. Or if you wish, authors can add a separate list of abbreviations.

For meta-analysis figures, please also add the software/databases used for analysis and figure preparation. It will help readers.

Author Response

We would like to thank all reviewers for the positive evaluation of our work; we appreciate it very much. We would also like to thank you for the comments and suggestions that significantly improved the manuscript. We believe that the edited version includes all the required information.

Response to reviewer 3:

3.1 Throughout the text, please replace resistance/resistant with tolerance/tolerant. Resistance/resistant usually refers to biotic stresses.

In the text, we are using the nomenclature from „Biochemistry & Molecular Biology of Plants„ Buchanan  et al., ISBN 9780470714218 where these terms are used also for abiotic stress responses. But we thank the reviewer for this note, it prompt us to check the terminology and we have detected some wrongly used terms.

3.2 In the introduction, authors can provide a little information about how a single or the combination of these abiotic stresses impacts agricultural output and food safety, most probably in the first paragraph.

We thank the reviewer for this suggestion which is similar to point 2.4 (reviewer 2). We have modified this part to make it more complex, as suggested.

3.3 In Table 1/2, and the main text and other parts, please also mention the botanical names within the parenthesis on the first mention.

We thank you for this suggestion. Tables 1 and 2 were modified.

3.4 The number of references is too high. Authors can replace some older references with new ones, and the same citation of some review articles can be repeated at different places instead of using the new reference(s) for basic statements. Here, I suggest some of the recently published articles regarding different abiotic stresses that need to be adjusted to provide up-to-date references such as doi:10.1111/plb.13510;  10.1093/jxb/erac073; 10.1002/tpg2.20279; 10.1111/tpj.15483; etc. Authors can also check some high-impact papers and please try to reduce some basic references with the above suggestions and use them multiple times.

We know that the number of our references is above average. On the other hand, this is significantly due to the large amount of data in the tables. We removed 12 references from the original list. Some references were redundant, some were replaced with fresh ones and new references were added as requested by other reviewers.

3.5 Please carefully check the whole text and define the undefined abbreviations on the first mention. Or if you wish, authors can add a separate list of abbreviations.

We would like to thank the reviewer for this gap in our text – we have checked the text as recommended.

3.6 For meta-analysis figures, please also add the software/databases used for analysis and figure preparation. It will help readers.

A similar comment was raised by reviewer 1. We have added the methodology as supplementary material 2.

Reviewer 4 Report

This review paper is nicely written. I just have one suggestion: if authors may compare Europe's situations to those of other continents, as shown in figure 1, it will be highly beneficial to readers.

Author Response

We would like to thank all reviewers for the positive evaluation of our work; we appreciate it very much. We would also like to thank you for the comments and suggestions that significantly improved the manuscript. We believe that the edited version includes all the required information.

Response to reviewer 4:

Reviewer 4)

4.1 This review paper is nicely written. I just have one suggestion: if authors may compare Europe's situations to those of other continents, as shown in figure 1, it will be highly beneficial to readers.

Dear reviewer, we understand your point. We have tried to prepare a text that will have a direct and logical organization from the beginning. For example, panel 1A is followed by publications focusing on major European crops, and this trend is further followed in the text. We would like to maintain this logic, and therefore, as an alternative, we have prepared additional information on abiotic stress from a global perspective at your suggestion. We believe that this change is acceptable from your point of view.

Round 2

Reviewer 3 Report

I am happy with the revised version and the authors responses. Thus, I can be considered for publication in IJMS.